# A revision of the rare *Strumigenys mnemosyne* (Formicidae; Myrmicinae) group using micro-CT scanning, with the description of three new species, and the virtual repair of a broken paratype

Matthew T. Hamer [1]*, Julian Katzke [2], Kit Lam Tang[1], André Ibáñez Weemaels[1], Francisco Hita-Garcia [2,3], Evan P. Economo[2,4], Benoit Guénard [1,5]

**1** School of Biological Sciences, The University of Hong Kong, Hong Kong SAR, China, **2** Biodiversity and Biocomplexity Unit, Okinawa Institute of Science and Technology Graduate University, Tancha, Onna-son, Okinawa, Japan, **3** Center for Integrative Biodiversity Discovery, Museum für Naturkunde, Berlin, Germany, **4** Department of Entomology, University of Maryland, College Park, Maryland, United States of America, **5** Hong Kong Biodiversity Museum, School of Biological Sciences, The University of Hong Kong, Hong Kong, China

* matt.hamer@hotmail.co.uk

## Abstract

The ant genus *Strumigenys* is both species rich, with over 800 species described, and morphologically diverse. The *Strumigenys mnemosyne* species group, a collection of small and infrequently collected *Strumigenys* known from across the Asian tropics and subtropics is revised. X-Ray microcomputed tomography alongside traditional microscopy is used describe three new species, *S. liui* **sp. nov.**, S. *marmorata* **sp. nov.**, and *S. rimdahli* **sp. nov.**, increasing the number of species within the group to eight. Further, micro-CT scans of the worker and gyne caste of *S. mazu* are produced, providing the first morphological description for the later caste. Owing to specimen damage obtained during specimen transportation for one of the type specimens used within this study, the utility of 3D models to reconstruct broken type material is explored. Updated distributions and dichotomous keys are also provided for the group.

## Introduction

The small, slow moving, and cryptobiotic ant genus *Strumigenys* is the third most species-rich ant genus with over 800 described species spread across the globe, with peaks of diversity observed within the tropical regions [1,2]. The genus is predominantly leaf litter dwelling, with few occupying the arboreal and subterranean stratums [3–5]). *Strumigenys* species have evolved an array of mandibular forms and ensnaring mechanisms, such as a latch mediated mandibular spring system used to

**Data availability statement:** All cybertype datasets, comprising CT scans (nii format), 3D models (STL format), shaded still image renders of specimen models (PNG & TIFF formats), and digital colour images can be accessed in Zenodo Digital Repository (https://zenodo.org/records/14830376). All 3D surface models generated within this study are available on Sketchfab (https://skfb.ly/p9yqI).

**Funding:** BG was awarded; Environment and Conservation Fund from the Government of the Hong Kong Special Administrative Region, under the 'Environmental Research, Technology Demonstration and Conference Projects' funding scheme, Project number ECF 137/2020. URL; https://www.ecf.gov.hk/en. Funder did not play any role in the study design, data collection and analysis, decision to publish, or preparation of the manuscript. MTH and AIW recieved salary from ECF 137/2020. BG was awarded; Research Grant Council from the Government of the Hong Kong Special Administrative Region, GRF17103223. URL https://www.ugc.edu.hk/eng/rgc/funding_opport/grf/. Funder did not play any role in the study design, data collection and analysis, decision to publish, or preparation of the manuscript. MTH recieved salary from GRF17103223.

**Competing interests:** The authors have declared that no competing interests exist.

capture soft-bodied, fast moving arthropod prey such as collembola [6]. The genus is morphologically diverse, not only in mandibular morphology, but also numerous morphological characters across the whole body, including (but not limited to) sculpture, pilosity and magnitude of spongiform tissue, particularly around the metasoma [3]. The genus currently consists of over 100 morphologically diverse species groups, diagnosed by a suite of morphological characters [3,7]. One such group, the *S. mnemosyne* species group, comprises members that are relatively small (often less than 2 mm in total length) and considered to be locally uncommon and rare across Asia [3] (Fig 1). Many species are known only from singleton holotype specimens, or only a handful of specimens. However, they are morphologically recognisable due to their highly reduced eyes (more often a single ommatidia), presence of a marginated elongate cuticular strip below the preocular carina, and reduced basal mandibular lamella [3].

X-Ray microcomputed tomography (micro-CT) is a powerful tool capable of producing 3-dimensional (3D) reconstructions of the whole organism or particular parts. Models of organisms can be comprehensively analysed within 3D virtual space via rotation, dissection (i.e., segmentation) and measurements. Although the application of micro-CT analyses are broad, in recent years researchers have used micro-CT data for taxonomic purposes, particularly for extinct arthropod taxa preserved in amber [8–10], but also extant taxonomic groups such as myriapods [11], spiders [12], beetles [13], as well as ants [14–19]. A notoriously challenging hurdle for systematists is access to the type material deposited in institutions across the world. Virtual collections of '*cybertypes*' can help to alleviate this hurdle, with 3D models of types stored online being considerably more accessible than traditional physical collections [16,19,20]. During specimen handling procedures, specimens can be broken purely by accident or by physical abrasion. This is particularly worrisome for high value specimens, such as type material. Micro-CT derived data could alleviate this hurdle by providing datasets of individually scanned body parts that can be 'repaired' in 3D virtual space.

Here micro-CT scanning image stills, 3D models, and traditional microscopy are used to review the *Strumigenys mnenosyne* group. Three novel species are described, *S. liui* **sp. nov**. from Yunnan, China, *S. marmorata* **sp. nov** from Hong Kong, and *S. rimdahli* **sp. nov.** from Thailand. A 3D model of a virtually repaired paratype specimen of *S. marmorata* that had been broken prior to scanning is also made available. For *S. mazu*, micro-CT scans are made available for both worker and queen. A novel description of the queen, as well as a new provincial record for Fujian Province, China is supplied. An updated and expanded species group definition is produced, incorporating novel morphological characters from *S. marmorata*. An updated dichotomous key is provided, alongside distributional checklist and map of the whole *S. mnenosyne* group, as well as high-resolution images of all species, including the first for *S. runa* [3].

## Materials and methods

### Sampling

Leaf litter sampling was conducted by members of the Insect Biodiversity and Biogeography Laboratory (IBBL) between 2014–2024 and by Dr. John Fellowes between

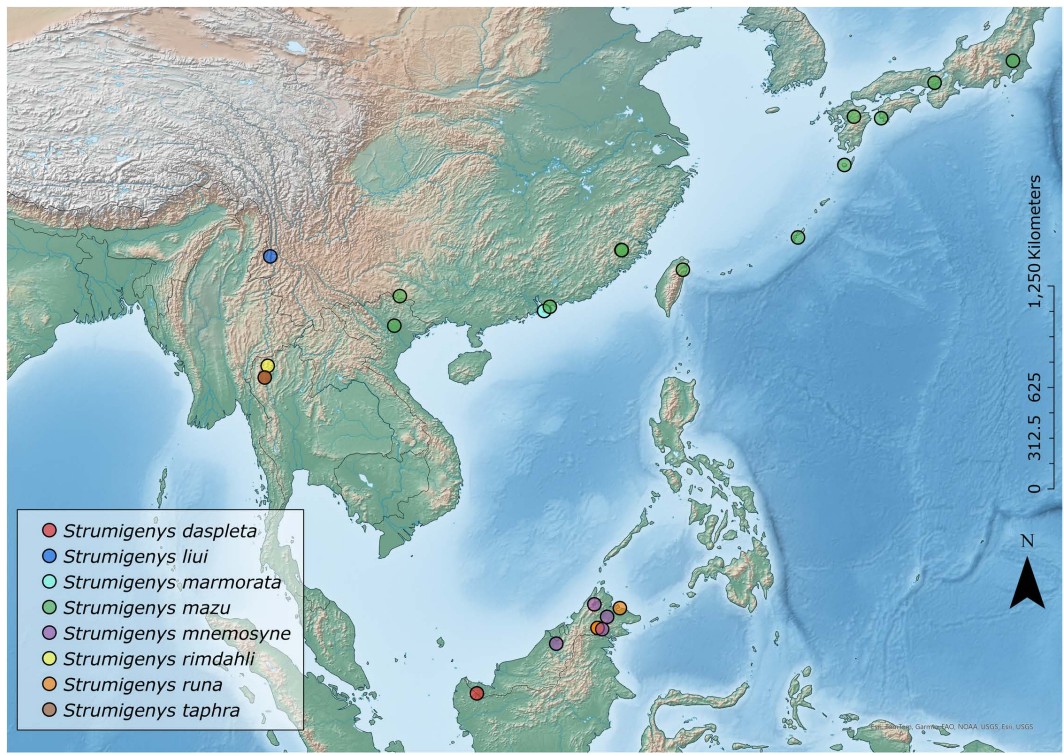

**Fig 1. Distribution map of the *Strumigenys mnemosyne* group across Asia.**

1993–2002. Specimens used in this study were primarily collected via leaf litter extraction, whereby leaf litter was sifted using a 5 mm meshed sifter and with filtrate extracted in a mini-Winkler between 3–7 days. Additional material was included within this study from expeditions to Hengduan Mountain range – Yunnan [21]), and Chiang Mai, Thailand. Confidently identified material made available on AntWeb.org were incorporated into this study, alongside examined specimens deposited in the Natural History Museum, London (NHMUK). The unidentifiable specimen from Java, mentioned by Bolton [1], was unfortunately not available for study.

## Measurements and images

Images and measurements were taken with a DMC5400 Camera attached to a Leica M205C Stereomicroscope and processed in Leica Application Suite (LASX). Measurements were accurate to 0.01 mm. Images were edited for artifacts using Adobe Photoshop, with plates made in Adobe InDesign. Measurements and indices followed Tang *et al* [22], and Brassard *et al* [5], with the addition of dorsal postpetiole length and dorsal postpetiole width. Sculpture terminology follows definitions outlined by [23].

- **HW** – Head Width: Maximum width of head in full-face view, not including eyes.

- **HL** – Head Length: Maximum length of head in full-face view, from anterior clypeal margin to midpoint of posterior head margin. If posterior margin concave, measurement is taken from a straight line between the anterior most points.

- **ML** – Mandible Length: Length of mandible in full-face view, from apex to anterior clypeal margin.

- **SL** – Scape Length: Length of scape from apical most point to basal constriction distal to antennal foramen.

- **WL** – Weber's Length: The diagonal length of the mesosoma from posterior basal angle of the metapleuron to the anterior most point of the pronotum.

- **PW** – Pronotum Width: Maximum width of the pronotum in dorsal view.

- **PL** – Petiolar Length: Maximum length of the petiole in lateral view, from the anterior most point of the petiole to posterior petiole margin. If propodeal lobes obscure the anterior most point, measurement was taken before the peduncle was obscured.

- **PH** – Petiolar Height: Maximum height of petiole from petiole node apex to base of ventral petiole margin perpendicular to PL. Measured in lateral view and ignoring spongiform tissue.

- **DPW** – Dorsal Petiolar Width: Maximum width of petiole in dorsal view, ignoring spongiform tissue.

- **DPL** – Dorsal Petiolar Length: Maximum length of petiolar node in dorsal view, ignoring spongiform tissue.

- **DPPW** – Dorsal Postpetiole Width: Maximum width of postpetiole in dorsal view, ignoring spongiform tissue.

- **DDPL** – Dorsal Postpetiole Length: Maximum length of postpetiole in dorsal view, ignoring spongiform tissue. If anterior margin concave, measurement is taken from a straight line between the anterior most points.

- **ATL** – Abdominal Tergum IV Length: maximum length of abdominal tergite four, measured in lateral view from anterior to posterior margin.

- **TL** – Total Length: The sum of ML, HL, WL, PL, DPPL and ATL.

**Indices**

**CI** – Cephalic index: $HW / HL \times 100$

**MI** – Mandibular index: $ML / HL \times 100$

**SI** – Scape index: $SL / HW \times 100$

**PI** – Pronotum index: $PW / HW \times 100$

**LPI** – Lateral Petiolar index: $PH / PL \times 100$

**DPI** – Dorsal Petiolar index: $DPW / PL \times 100$

**DPPI** – Dorsal Post-petiolar index: $DPPW / DPPL \times 100$

**Depository institution abbreviations**

- **ANIC** – The Australian National Insect Collection, Canberra, Australia

- **NHMUK** – The Natural History Museum, London, United Kingdom.

- **HKBM** – Hong Kong Biodiversity Museum, School of Biological Sciences, The University of Hong Kong

- **IBBL** – Insect Biodiversity and Biogeography Laboratory (IBBL), School of Biological Sciences, The University of Hong Kong

- **JTLC** – John T. Longino Collection. University of Utah. Salt Lake City, Utah. USA

- **MCZC** – Museum of Comparative Zoology, Harvard University, Cambridge, Massachusetts, USA

- **MHNG** – Museum of Natural History, Geneva, Switzerland

- **MNHN** – Muséum National d'Histoire Naturelle, Paris, France.

- **ZRC** – Zoological Reference Collection, Lee Kong Chian Natural History Museum, Singapore

### X-ray micro-CT scanning and visualization

Ethanol preserved specimens were stained using 4% iodine dissolved in pure ethanol solution for at least three days prior to micro-CT scanning. For scanning, specimens were thoroughly washed and transferred into ethanol-filled and then sealed plastic pipette tips that were clamped into the scanner's specimen holder. Pinned specimens were directly clamped into the holder via the paper tip they are glued on. The broken paratype of *S. marmorata* was also scanned dry in a sealed pipette tip. Specimens were scanned at the Okinawa Institute of Science and Technology Graduate University (Japan) with a Zeiss Xradia 510 Versa laboratory micro-CT scanner employing Zeiss's Scout-and-Scan Control System Reconstructor (v. 11.1.6411.17883) for tomographic reconstruction. Scan parameters were set individually to optimise resolution and contrast depending on preservation method and body size. 3D model reconstruction was conducted in Amira (v. 2020.2, Thermo Fisher Scientific) using a workflow to create a detailed surface mesh of the exterior morphology of the *Strumigenys* specimens. First, all voxels were threshold-segmented corresponding to an ant within a scan and unwanted segmented loose contaminants were subsequently excluded by largest-island selection. All remaining internal voxels were segmented using Amira's 'Compute Ambient Occlusion' module. These voxels were locked and the original threshold dilated to create a mask for the scan. This mask was multiplied with the original scan using an 'Arithmetic' module to create a new 3D image keeping the dilated-threshold regions as in the original scan but assigning all internal voxels to a maximum value. Thresholding this new image essentially keeps all exterior detail but omits all interior detail as desired for visualisations and file-size reduction. Using the 'Isosurface' module, thresholded surface meshes were exported as STL files for rendering in Blender (v. 3.6, Blender Foundation). In Blender, simple studio-lighting conditions with four light sources were created, with models rendered to include ambient occlusion pass to highlight details. Scale bars were included in all renders.

### Cybertypes

All cybertype datasets, comprising CT scans (nii format), 3D models (STL format), and shaded still image renders of specimen models (PNG & TIFF formats), can be accessed in Zenodo Digital Repository (https://zenodo.org/records/14830376). Digital colour images of each newly described species are also provided (TIFF & PNG formats). Further, following Hita Garcia *et al* [19], all 3D surface models generated within this study are available on Sketchfab (https://skfb.ly/p9yql).

### Species concept and species delimitation

Owing to the lack of material across the species group, no molecular data is included in this study. The available number of specimens for each species, other than *Strumigenys mazu*, was very low, often less than three individuals, and in some cases only one. Further, many specimens that were examined were owned by private collections or museums which did not permit DNA extractions. Species are delimited using sets of detailed discrete morphological characters frequently used to delineate species boundaries in previous *Strumigenys* taxonomic work [3,22,24]. Despite low numbers of type material for some species, our morphological character sets strongly support our species hypotheses, making our newly described *Strumigenys* species recognisable as unique taxonomic entities.

### Nomenclatural acts

The electronic edition of this article conforms to the requirements of the amended International Code of Zoological Nomenclature, and hence the new names contained herein are available under that Code from the electronic edition of

this article. This published work and the nomenclatural acts it contains have been registered in ZooBank, the online registration system for the ICZN. The ZooBank LSIDs (Life Science Identifiers) can be resolved and the associated information viewed through any standard web browser by appending the LSID to the prefix "http://zoobank.org/". The LSID for this publication is: urn:lsid:zoobank.org:pub:5A5A90F8-4609-4C9C-9453–2CF8E9EE3E76. The electronic edition of this work was published in a journal with an ISSN, and has been archived and is available from the following digital repositories: PubMed Central, LOCKSS.

## Results

### Updated *mnemosyne* species group diagnosis

Adapted from Bolton [3]

- Mandibles in full-face view and at full closure triangular, the masticatory margins engage throughout their visible length and are serially dentate. In ventral view outer margin of mandible without a prebasal inflected angle. MI 15–19.

- Dentition. Basally with a dental row of 5 triangular teeth, first tooth sometimes smaller than subsequent teeth, remaining teeth are about equal in size. These are immediately followed by two slightly smaller teeth (about the size of the first tooth), followed by 4 denticles (of which the basal most two are somewhat larger than the apical most two) and a small to medium sized apical tooth, giving a total dental count of 12.

- Basal lamella minute to vestigial, at most a mere ridge on the margin proximal of the basal tooth, not visible in full-face view when the mandibles are fully closed.

- Labrum terminates in a pair of narrow triangular to digitate lobes.

- Clypeus with anterior margin subtlety convex to transverse to shallowly concave across its entire width.

- Sides of clypeus convergent anteriorly.

- Clypeal dorsum with minute appressed pubescence or pubescence plus erect short simple setae; clypeal margins with or without freely projecting setae.

- Preocular carina short and narrow in full-face view, usually only its lateral edge visible from posterior margin of clypeus to level posterior end of frontal lobe.

- Side of head immediately below preocular carina with an elongate narrow depressed area of cuticle that is of different colour or texture from the surrounding cuticle and may be distinctly margined; the area may have a glandular function.

- Ventrolateral margin of head between eye and mandible rounded to angular. Postbuccal impression deep and conspicuous.

- Cuticle of side of head within the scrobe area smooth and shining.

- Scape short, SI 50–63, not strongly dorsoventrally flattened and without a flange-like leading edge.

- Leading edge of scape usually with 2–3 straight simple setae that project anteriorly or anterodorsally; or without projecting setae.

- Pronotal dorsum without a median longitudinal carina.

- Propodeum without trace of spines or teeth, the declivity on each side with a moderate to broad conspicuous lamella.

- Spongiform appendages of waist segments massively developed in profile, but in dorsal view the posterior collar of the petiole node vestigial or absent. Base of first gastral sternite in profile with spongiform tissue feebly developed or absent.

- Pilosity. Pronotal humeral seta present, simple, longer than others on the dorsum but not strongly differentiated from them. Head either with highly appressed setae or with simple standing setae that project freely from the dorsal surfaces of the head, mesosoma, waist segments and gaster. Freely projecting elongate simple setae present or absent on the dorsolateral margins of the head, the dorsal (outer) surfaces of the middle and hind tibiae, and the outer surfaces of the basitarsi.

- Sculpture. Dorsum and sides of the mesosoma, declivity of propodeum, waist segments and gaster all smooth and shining; head may also be smooth or have reticulate-punctate sculpture.

**X-ray micro-CT results**

Micro-CT data for six *Strumigenys* specimens ranging from 0.65 to 1.35 µm voxel size in spatial resolution are provided. Original scans were cropped to reduce file sizes and improve computational performance and provide the scans as stacks of PNG files. To correctly display these scans, voxel size or spacing must be adjusted based on the provided metadata (https://zenodo.org/records/14830376). Further, 3D models are provided as surface meshes in STL format representing the models used for rendering, which already incorporate correct scaling information in micrometers.

**Taxonomic accounts**

**_Strumigenys daspleta_** (Bolton, 2000)
   *Pyramica daspleta* Bolton, 2000: 445 (w.) BORNEO (East Malaysia: Sarawak).
   Combination in *Strumigenys*: Baroni Urbani & De Andrade, 2007: 118. Fig 2A–C.

**Diagnostic characters**

*Strumigenys daspleta* can be distinguished from other members of the *S. mnemosyne* group by the pre-ocular blister-like cuticle on the posterior lobes, the numerous standing setae on the cephalic dorsum and the dorsolateral margin of the head, in full-face view, with 6 or more freely projecting setae (among other characters) [3].

**Material examined**

Specimen images examined ($n=1$):
   **Holotype** worker: Malaysia; Sarawak, Gn Penrissen. 1000m asl. 23 May 1994 Coll. Lobl Burckhardt. Det. Barry Bolton. CASENT0102600 [MHNG]

**Distribution**

Only known from East Malaysia (Sarawak) (Fig 1; Table 1).

**Comments**

As is characteristic of the species group, *S. daspleta* is only known from a single record. The holotype specimen was collected from '*Gn Penrissen*' Sarawak, Malaysia. The type locality is likely Gunong (mount) Penrissen located in the west of Borneo. Nothing is known of the biology of *S. daspleta*.

**_Strumigenys liui sp. nov._** (Hamer, Tang & Guénard, 2025)
   urn:lsid:zoobank.org:act:93EF59CF-F5B6-49E0-9211–7CC64D725EF9 Fig 3A–F.

**Diagnosis**

In full-face view, head distinctly longer than wide; cephalic dorsum entirely smooth. Clypeal dorsum with short, simple, appressed setae directed anteromedially; pair of standing setae isolated to posterior margin. In lateral view, posterior

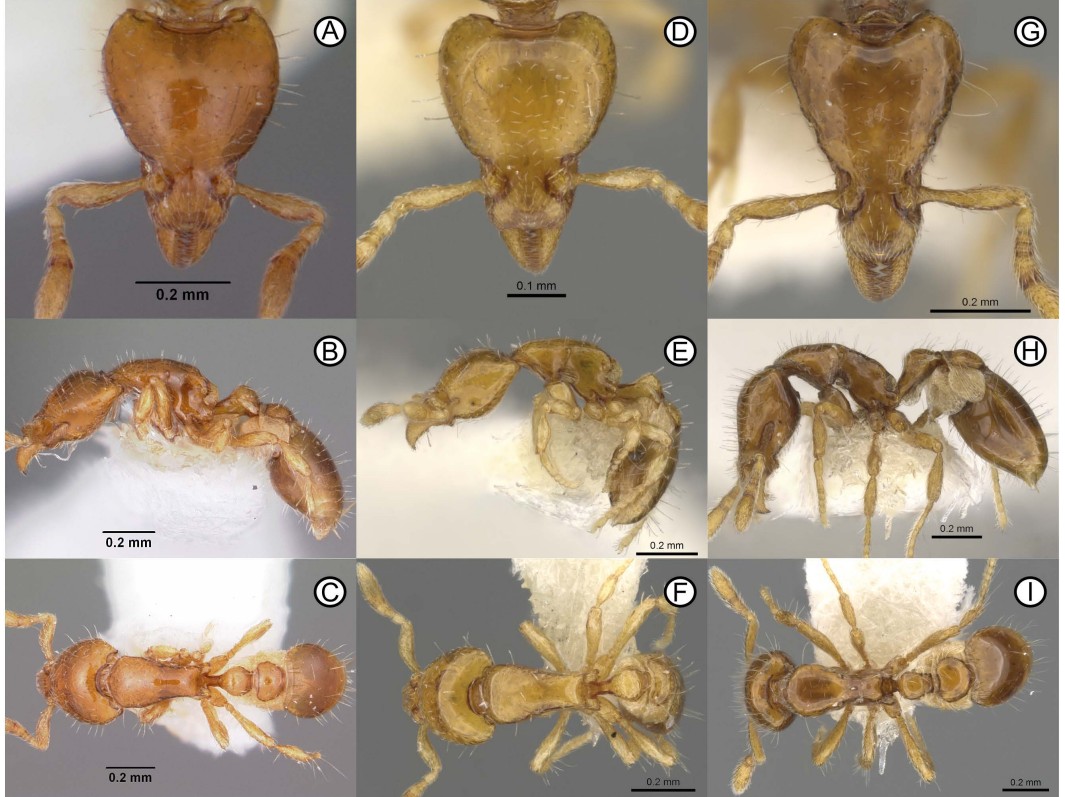

**Fig 2. Type specimens of the *Strumigenys mnemosyne* group.** Holotype specimen of *Strumigenys daspleta* (CASENT0102600, photographer Michele Esposito) (A-C), holotype specimen of *S. mnemosyne* (CASENT0900146, photographer Michele Esposito) (D-F), paratype specimens of *S. taphra* (CASENT0900147, photographer Michele Esposito) (G-I). Images available from antweb.org.

**Table 1. Updated *Strumigenys mnemosyne* group distributional checklist.**

| Species | China; Fujian Prov. | China; Guangxi Prov. | China; Hong Kong SAR | China; Yunnan Prov. | Taiwan | Japan | East Malaysia (Borneo) | Thailand | Vietnam |
|---|---|---|---|---|---|---|---|---|---|
| *Strumigenys daspleta* (Bolton, 2000) | | | | | | | ✓ | | |
| *Strumigenys liui* **sp. nov.** | | | | ✓ | | | | | |
| *Strumigenys marmorata* **sp. nov.** | | | ✓ | | | | | | |
| *Strumigenys mazu* (Terayama et al., 1996) | ✓ | ✓ | ✓ | | ✓ | ✓ | | | ✓ |
| *Strumigenys mnemosyne* (Bolton, 2000) | | | | | | | ✓ | | |
| *Strumigenys rimdahli* **sp. nov.** | | | | | | | | ✓ | |
| *Strumigenys runa* (Bolton, 2000) | | | | | | | ✓ | | |
| *Strumigenys taphra* (Bolton, 2000) | | | | | | | | ✓ | |
| **Total** | 1 | 1 | 2 | 1 | 1 | 1 | 3 | 2 | 1 |

margin of glandular trench clearly abutting eye. Katepisternum with long anteriorly directed free cuticular spur reaching posteroventral corner of pronotal side. Petiolar node with distinct anterior face in lateral view. Lateral margins of petiolar disc converging with anterior margin in parabolic arc in dorsal view. Postpetiolar disc in dorsal view distinctly wider than long. Basigastral costulae absent.

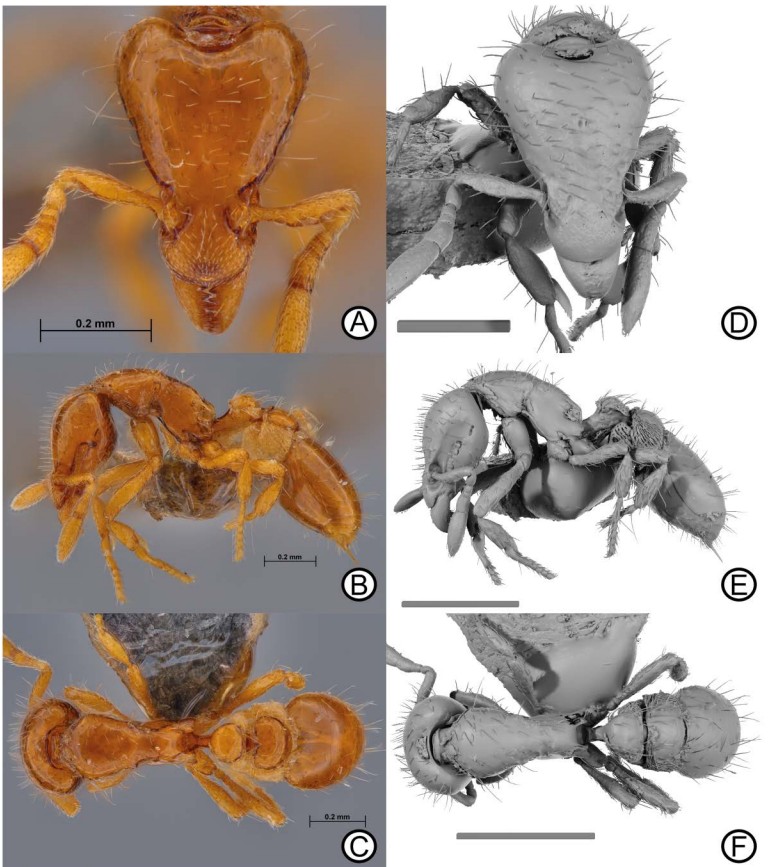

**Fig 3. Holotype specimen of *Strumigenys liui* sp. nov. worker (MCZ-ENT00759758).** Digital images (A) head, (B) lateral, (C) dorsal. Micro-CT scan stills (D) head, (E) lateral, (F) dorsal. Scale bars = 0.2 mm (D), 0.5 mm (E, F).

## Material examined

Physical specimen examined (*n* = 1):

   **Holotype** worker: China; Yunnan: Hengduan Mountain, Baihua Ling; N25.302, E98.789; 1900 m a.s.l; 3 July 2019; Forest, sifted litter, C. Liu #340. MCZ-ENT00759758. ANTWEB1010117 [MCZC].

## Cybertype

Holotype (MCZ-ENT00759758) dataset consists of surface volume rendering image stacks of PNG files comprising full-face views stills of the head, whole body in lateral and dorsal view, as well as dorsal view of postpetiole and first gastral tergites. Further, raw CT-scan data (nii format), and a full 3D surface model (STL format) are made available. Digital stacked colour images (tif format) of the head in full-face view, and the whole body in dorsal and lateral view are also provided. All data is deposited on Zenodo (https://zenodo.org/records/14830376). A 3D surface model is also made available on Sketchfab (https://skfb.ly/pyFxD).

## Description

Holotype worker: HW 0.35; HL 0.48; ML 0.09; SL 0.21; WL 0.52; PW 0.23; PL 0.24; PH 0.13; DPW 0.14; DPL 0.11; DPPW 0.19; DPPL 0.13; ATL 0.42; TL 1.85.

   Indices: CI 74.06; MI 19.25; SI 59.04; PI 64.69; LPI 55.93; DPI 58.05; DPPI 148.44.

## Head

In full face-view, head longer than broad, broadest at posterior lobes. Posterior margin deeply concave, with thin lamellae extending across margin width; granulate cuticular structure present anteriorly to inter-posterior margin lamellae. Lateral clypeal margins converging anteriorly; anterior margin convex. Epistomal sulcus absent, cephalic dorsum and clypeus continuous; torular lobe extensive, weakly convex and obscuring antennal foramen in full-face view. Antennae totalling six segments; scape short, failing to reach posterior margin; scape weakly dorsoventrally flattened, converging anteriorly and forming distinct carinae at leading edge, narrowing basally and distally. Flagellomeres terminating in two-segment club; apicalmost segment distinctly longer than penultimate segment. Mandible triangular, reducing in width apically; basal masticatory margin with five long, triangular teeth, apical half with two shorter teeth, followed by four small teeth and apical tooth slightly larger than previous four denticles. In lateral view, preocular carina short, terminating long before eye level. Scrobe absent. Eye minute, comprised of two ommatidia. Glandular groove well impressed, extending back to eye, margins of both structures abutting. Post-buccal impression conspicuous. Ventral head margin, from preocular notch to occipital foramen, weakly sinuate being convex before eye and concave anterior to occipital foramen. In lateral view, angle of intercept between ventrolateral head margin and posterior head margin obtuse and blunt.

## Mesosoma

In dorsal view, mesosoma lacking promesonotal suture, metanotal groove absent In lateral view, promesonotal suture weakly impressed, not reaching dorsolateral margin. Mesosomal outline weakly convex anteriorly to strongly convex posteriorly due to propodeal declivity. Dorsolateral margin rounded, lacking margination. Mesopleural notch present and well impressed. Hair wheel present. Katepistermum with long, sharp, anteriorly directed cuticular spur reaching posteroventral corner of pronotal side. Propodeal spiracle rounded, located high on propodeum. Metapleural gland bulla teardrop shape, anterodorsal corner terminating in acute angle. In dorsal view, pronotum wider than long, reaching maximum width anteriorly; lateral margins converging anteriorly with anterior margin convex. Propodeum with wide, semi-rectangular, rounded spongiform tissue. Propodeal declivity slightly concave in dorsal view.

## Metasoma

In lateral view, petiolar peduncle short, node with distinct anterior face; dorsal nodal surface convex. Petiolar spiracle located just below node on anterolateral region of petiole. In dorsal view, petiolar disc wider than long, lateral margins converge with anterior margin in parabolic arc, posterior margin straight; postpetiolar disc wider than long, anterior margin feebly concave, lateral and posterior margins convex. In lateral view, spongiform tissue curtain present on peduncle ventrally, and on node posterodorsally. Spongiform tissue massively produced around postpetiole ventrally and laterally. First gastral tergite long, as long as petiole and postpetiole combined; in dorsal view, anterolateral corners of first gastral tergite gradually curving; anterior region of tergite with spongiform tissue.

## Pilosity

Clypeus with short, simple, appressed setae directed anteromedially; pair of standing setae present on posterior clypeal margin, standing setae absent across remaining clypeal dorsum. Antennal segments with appressed long setae; leading margin of scape with at least two projecting setae. Mandibular dorsum with small, firmly appressed setae. Cephalic dorsum with numerous, medium length, erect, apically acute setae; few appressed longer setae also present, directed anteromedially; ventral surface of head with subdecumbent and suberect, medium length setae. In full-face view, seven long, erect simple setae located along upper scrobe margin arranged serially and directed anterolaterally. Mesosomal dorsum with scattered long, erect and decumbent setae across dorsum; fewer setae on propodeal dorsum, declivity glabrous. Pronotal humeral seta long, erect, longer than surrounding setae and directed posterolaterally. Lateral mesosomal surface almost glabrous, some setae present on pronotal dorsolateral margin zone; mesopleuron and lateral propodeal surface

glabrous. Petiolar dorsum with subdecumbent, long and medium length setae, all directed posteriorly. Postpetiolar dorsum with long, erect setae. Dorsum of gastral tergite one with numerous long erect setae scattered across the surface. Gastral sternites with appressed medium length setae, and many short, erect setae. Coxa, trochanter, femur and tibia with numerous subdecumbent setae, as well as scattered long erect setae particularly on outer surface of metatibia and basitarsus.

## Sculpture

Mandibles and antennal segments dulled by dense micropuncture. Entire body integument lacking sculpture, smooth. Basigastral costulae on first gastral tergite entirely absent.

## Colour

Body reddish orange, appendages yellow. Spongiform tissue light yellow.

## Etymology

Named after our colleague and friend, Dr. Cong Liu, in recognition of his ecological and taxonomic work on ants from China and other regions.

## Distribution

Only known from Yunnan, China (Fig 1; Table 1).

## Comments

Although only one individual was collected and described, adequate morphological evidence is provided to separate *S. liui* from its closest morphological relative, *S. taphra*. Further holotype images, and paratype specimens of *S. taphra* were examined and compared with images and CT-scans of *S. liui*. Morphological differences of particular note are the alternative pilosity on the clypeal dorsum being, short, highly appressed, instead of longer and more filamentous. Further, the absence of any basigastral costulae on the first gastral tergite, as well as the entirely smooth head dorsum (smooth and punctuate in *S. taphra*) further differentiates each species. The lack of additional material is not atypical for this group of *Strumigenys*. In the past, several species from the same group have been described from single individuals (e.g., *S. daspleta, S. mnemosyne* & *S. runa*). Thus, given the strong differing characters provided here, and the past acceptance of holotype only descriptions, it can be deemed acceptable to describe *S. liui* as a new species based on a single individual. As this specimen was previously identified as *S. taphra*, all mentions of this specimen should be revised to reflect the new taxonomy provided here.

## Ecology

Only known from a single individual from Hengduan Mountain range in Yunnan, China. The specimen was obtained via sifted leaf litter, and like *S. taphra*, at high elevation (1900 m a.s.l), suggesting tolerance for higher elevation conditions.

***Strumigenys marmorata sp. nov.*** (Hamer, Tang & Guénard, 2025)
   urn:lsid:zoobank.org:act:01240912–0FCD-42C9-8FC2–70FC6FB4F06E Fig 4A–F, Fig 5.

## Diagnosis

In full-face view, head longer than broad. Standing setae absent on cephalic dorsum in lateral view as well as dorso-lateral margin. Clypeal dorsum with short, narrowly spatulate, appressed setae, directed toward anterior clypeal margin. Basal masticatory margin with five long, triangular teeth, apical half with two shorter teeth, followed by four small teeth and apical tooth larger than previous four denticles. Scape without standing setae, appressed simple setae only. Scape dorsoventrally flattened, equal in width across length. Mesopleural notch impressed and hair wheel present. Mesosoma

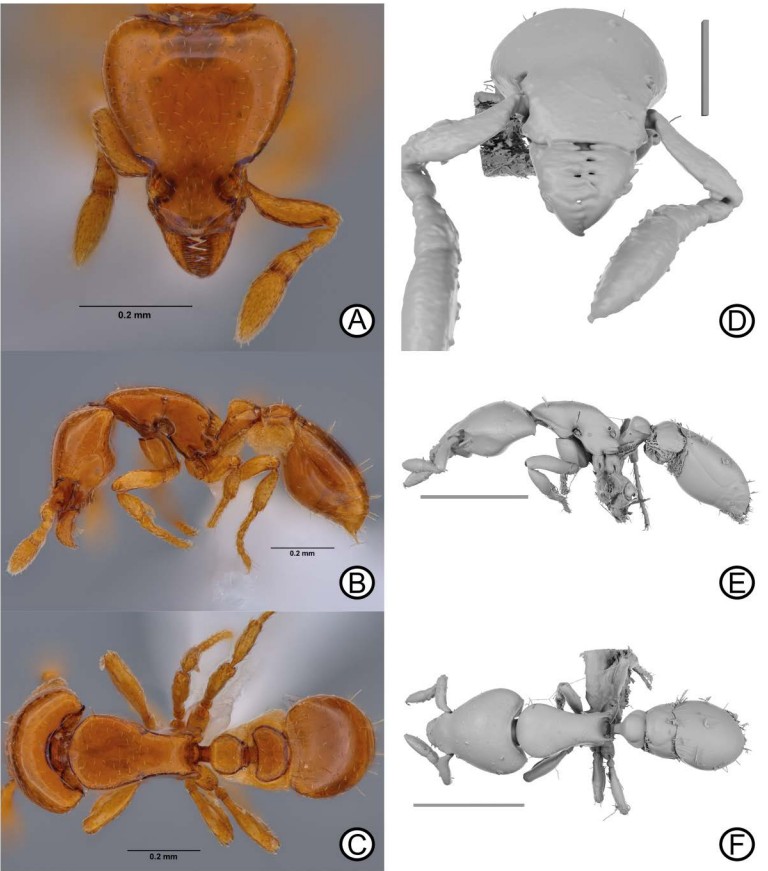

**Fig 4. _Strumigenys marmorata_ sp. nov. holotype specimen (ANTWEB1010070) and paratype specimen (CASENT0764984).** Digital images of holotype (A) head, (B) lateral, (C) dorsal. Micro-CT scan stills of _S. marmorata_ paratype specimen (CASENT0764984) (D) head, (E) lateral, (F) dorsal, scale bars = 0.2 mm (D), 0.5 mm (E, F).

lacking standing setae, appressed short setae only. Meso- and metatibia lacking projecting seta. Pronotal humeral setae erect, stout and short, apically blunt. Integument lacking sculpture across body, other than basigastral costulae, coxae and basitarsi.

## Material examined

Physical specimens examined (_n_ = 2):

**Holotype** worker; China; Hong Kong SAR, Lantau; Shek Mun Kap; 22.26906 113.93088; 22 m a.s.l; 26 November 2022; Coll. H.C. Yin; 1m² leaf litter sample; disturbed secondary forest. ANTWEB1010070 [OSFMK-P2-01] [ZRC].

1 **Paratype** worker; China; Hong Kong SAR, Lok Ma Chau, Fung Kung, 22.51211 114.0928; 48 m a.s.l; 18 May 2022; Coll. A.I.Weemaels & C. Neeves; 0.5 m² leaf litter sample; disturbed forest. CASENT0764984 [FK1T1W1-5] [HKBM].

The paratype specimen broke into three pieces whilst opening the specimen transportation box (Fig 5). The mesosoma remains attached to the mounting point, whilst the head and metasoma are within 75% ethanol.

## Cybertype

The paratype (CASENT0764984) dataset consists of surface volume rendering image stacks of PNG files comprising isolated stills of the head in lateral view, the mandibles in anteroventral view, the metasoma and petiole in lateral view,

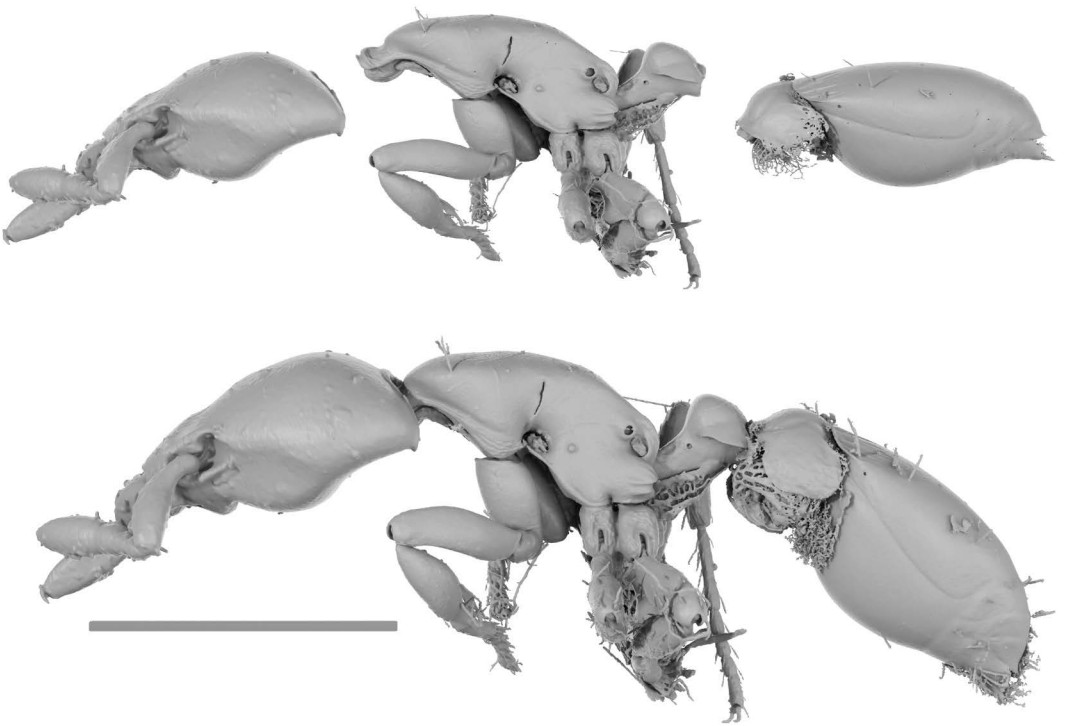

**Fig 5. Reconstructed paratype of *Strumigenys marmorata* sp. nov. (CASENT0764984).** A digital reconstruction of the paratype specimen of *Strumigenys marmorata* sp. nov. Scale bar = 0.5 mm.

and finally the postpetiole and gaster in lateral view. Additional surface volume renders include the whole reconstructed body in lateral and dorsal view. Further, raw volumetric data (nii format), and a full 3D surface model (STL format) are made available. Digital stacked colour images of the head in full-face view, and whole body in dorsal and lateral view are also made available of the holotype specimen (ANTWEB1010070) (TIF format), as well as the paratype specimen (CASENT0764984) (PNG format), with images of the later acquired prior to breakage. All data is deposited on Zenodo (https://zenodo.org/records/14830376). A fully reconstructed paratype (CASENT0764984) 3D surface model of the whole body is also made available on Sketchfab (https://skfb.ly/pyFxF).

## Description

Holotype measurements: HW 0.34; HL 0.39; ML 0.07; SL 0.17; WL 0.40; PW 0.21; PL 0.18; PH 0.11; DPW 0.11; DPL 0.10; DPPW 0.15; DPPL 0.09; ATL 0.31; TL 1.45.

  Indices: CI 87.18; MI 18.97; SI 50.59; PI 60.59; LPI 60.11; DPI 60.11; DPPI 163.44.

  Paratype measurements (*n* = 1): HW 0.39; HL 0.44; ML 0.08; SL 0.20; WL 0.46; PW 0.24; PL 0.21; PH 0.10; DPW 0.125; DPL 0.089; DPPW 0.169: DPPL 0.114; ATL 0.408; TL 1.68.

  Indices: CI 89.1; MI 18.4; SI 50.6; PI 61.8; LPI 48.8; DPI 60.4; DPPI 148.25.

## Head

In full face-view, head longer than broad, broadest anterior to posterior margin. Posterior margin broadly concave. Lateral clypeal margins converging anteriorly; anterior margin straight to subtly convex. Epistomal sulcus absent, cephalic dorsum and clypeus continuous; torular lobe small, convex and obscuring antennal foramen in full-face view. Antennae totalling six

segments; scape short, failing to reach posterior margin; scape dorsoventrally flattened, converging anteriorly and forming flange at leading edge, narrowing basally and distally. Flagellomeres terminating in two-segmented club; apicalmost segment longer than penultimate segment. Mandible triangular, reducing in width apically; basal masticatory margin with five long, triangular teeth, apical half with two shorter teeth, followed by four small teeth and apical tooth larger than previous four denticles. In lateral view, preocular carina short, terminating long before eye. Scape not well impressed; integument smooth. Eye minute, comprised of single ommatidium. Postbuccal impression conspicuous; ventral head margin, from preocular notch to occipital foramen, significantly sinuate being convex before eye then concave anterior to occipital foramen. In lateral view, angle of intercept between ventrolateral head margin and posterior head margin obtuse, with posterior projecting cuticular lobe at intercept.

## Mesosoma

In dorsal view, mesosoma lacking promesonotal suture, metanotal groove absent In lateral view, promesonotal suture weakly impressed, not reaching dorsolateral margin. Mesosomal outline weakly convex anteriorly to strongly convex posteriorly due to propodeal declivity. Dorsolateral margin rounded, lacking margination. Mesopleural notch present and well impressed. Hair wheel present. Katepistermum with long, sharp, anteriorly directed cuticular spur almost reaching posteroventral corner of the pronotal side. In dorsal view, pronotum evenly rounded anteriorly, reaching maximum width at mid-length. Propodeum unarmed; propodeum with semi-rectangular, dorsally rounded spongiform tissue only. Propodeal declivity slightly concave in dorsal view.

## Metasoma

In lateral view, petiole clavate; petiolar node slightly longer than wide; peduncle and anterior petiolar face meeting at conspicuous obtuse angle; anterior face of petiole flat, grading into dorsal face with curved angle. Dorsal margin of petiole and postpetiole surface convex. In dorsal view, petiole oval, wider than long. In dorsal view, postpetiole reniform, anterior margin conspicuously concave, lateral margins converging in parabolic arc posteriorly, posterior margin with concave notch medially. Ventral petiolar spongiform tissue present as short curtain; lateral petiolar spongiform tissue comparatively small. Conspicuous ventral and lateral spongiform tissue on postpetiole, obscuring lateral surface. Spongiform tissue surrounding postpetiolar disc anteriorly and posteriorly in dorsal view, but not on top of postpetiolar disc itself.

## Pilosity

In lateral view, cephalic dorsum and dorsolateral margins lacking standing setae. Setae on cephalic dorsum in full-face view short and appressed. Spatulate setae concentrated anteriorly on cephalic dorsum, directed inwards to cephalic mid-line. Simple setae concentrated on posterior cephalic dorsum, extending over posterior lobes and onto lateral cephalic surface. Dorsum of clypeus lacking standing setae, setae instead narrowly spatulate, appressed and directed anteriorly. Setae on both cephalic and clypeal dorsa well-spaced. Ventral cephalic setae mix of short, appressed setae alongside long, appressed simple setae. Postbuccal cavity with conspicuous series of long and erect simple setae noticeably longer than surrounding setae on cephalic dorsum. Scape lacking standing setae, appressed, simple setae only. Flagellomere segments with short, appressed simple setae, reaching highest concentration on apical segment. Pronotal humeral seta conspicuous, simple and apically blunt, directed antero-laterally and well differentiated from surrounding mesosomal setae. Dorsal mesosomal setae appressed and simple, directed inwards to mesosomal longitudinal midline. Hair wheel with three to five, appressed simple setae. Petiole and postpetiole disc with same seta type, but directed in random directions. Disc of postpetiole also with erect simple setae, with one pair located on the posterior margin, directed posterodorsally. Gastral tergites with both stout erect setae and short appressed setae. Gastral sternites in lateral view, with long, erect simple setae, noticeably shorter and less stout than those on gastral dorsum. Gastral sternites also with long, appressed to semi-erect simple setae. Dorsal (outer) surfaces of meso- and meta-basitarsi and tibiae lacking long projecting setae; short, decumbent setae on tibiae and tarsal segments only; setae directed towards respective segment apex.

## Colour

Body colour reddish brown, appendages light reddish brown, spongiform tissue light yellow.

## Sculpture

Almost entire body integument lacking sculpture, the only sculpture located on first gastral tergite in the form of basigastral costulae, and micro-punctures on antennae, coxae and basitarsi. Integument reflecting under non-diffused light.

## Etymology

From Latin marmorāta *f* ("marbled, covered or encrusted with marble"), nominative feminine singular of marmorātus. Named for the highly smooth integument surface across the body which resembles marble, a highly smooth, shiny rock when cut and polished.

## Distribution

Only known from Hong Kong, China (Fig 1; Table 1).

## Comments

*Strumigenys marmorata* sp. nov is a member of the *S. mnemosyne* group sharing all characters of the species group [3]. However, interesting morphological differences are apparent between *S. marmorata* and other members of the species group. These include seta differences on scape, and a lack of setae across the body. These characters have been integrated into the species group definition.

Within the *S. mnemosyne* group, *S. marmorata* most closely resembles *S. mnemosyne* and *S. rimdahli* due to the lack of standing setae on the scape. *Strumigenys marmorata* can be differentiated from *S. mnemosyne* by the lack of standing setae on the cephalic, mesosomal and petiole dorsum. Abrasion is unlikely here due to the lack of setae bearing points embedded within the integument of *S. marmorata*. Clypeal setae in *S. marmorata* are similarly arranged as in *S. mnemosyne* but are narrowly spatulate in *S. marmorata*. The scape in *S. mnemosyne* is constricted proximally but in *S. marmorata* the scape is as wide proximally as apically. The head width of *S. marmorata* is distinctly wider than in *S. mnemosyne* (0.39 and 0.30 mm respectively), and in fact is the widest in the species group, even wider than *S. taphra* (HW 0.35–0.37 mm), the largest member of the *S. mnemosyne* group (TL 1.9–2.0). *Strumigenys mnemosyne* was collected by Bolton in Sarawak (Malaysia), and is known from other localities all restricted to Borneo Island, > 1650 km from the collection locality in Hong Kong. Given the large geographic distance and unique morphological characters, misidentification between *S. marmorata* and *S. mnemosyne* is unlikely. Interestingly, the smooth and shining cuticular surface (including the femora and tibia) is a rare characteristic within *Strumigenys* species [7]. *Strumigenys marmorata* is also morphologically similar to *S. rimdahli*, however both can be readily separated by the presence of a sharp dorsolateral mesosomal margin in *S. rimdahli* (rounded in *S. marmorata*), and a convex rather than flat mesosomal dorsum in *S. marmorata*.

## Ecology

*Strumigenys marmorata* is known from two individuals collected within two distinct leaf litter samples from lowland areas in Hong Kong. At the paratype collection locality, an additional 24 leaf-litter samples were carried out to obtain additional individuals, but none were collected unfortunately, suggesting the rarity of this species. Interestingly, both holotype and paratype collection localities had numerous tramp and exotic ant species, including several *Strumigenys* species such as, *Strumigenys emmae* (Emery, 1890), *S. membranifera* Emery, 1869 and *S. nepalensis* De Andrade, 1994. Other tramp and exotic ant species included, *Tetramorium lanuginosum* Mayr, 1870, *Brachyponera obscurans* (Walker, 1859), *Pheidole megacephala* (Fabricius, 1793) and *Ooceraea biroi* (Forel, 1907) were collected at each site.

Collection sites were far from pristine, with surface leaf litter light, heavily composed of loose soil and general anthropogenic detritus.

***Strumigenys mazu*** (Terayama, Lin & Wu, 1996)

 *Smithistruma mazu* Terayama et al., 1996: 337, figs. 26, 27, 30, 31 (w.) TAIWAN.

 Combination in *Pyramica*: Bolton, 1999: 1673. Combination in *Strumigenys*: Baroni Urbani & De Andrade, 2007: 123. Fig 6A–F, Fig 7A–F.

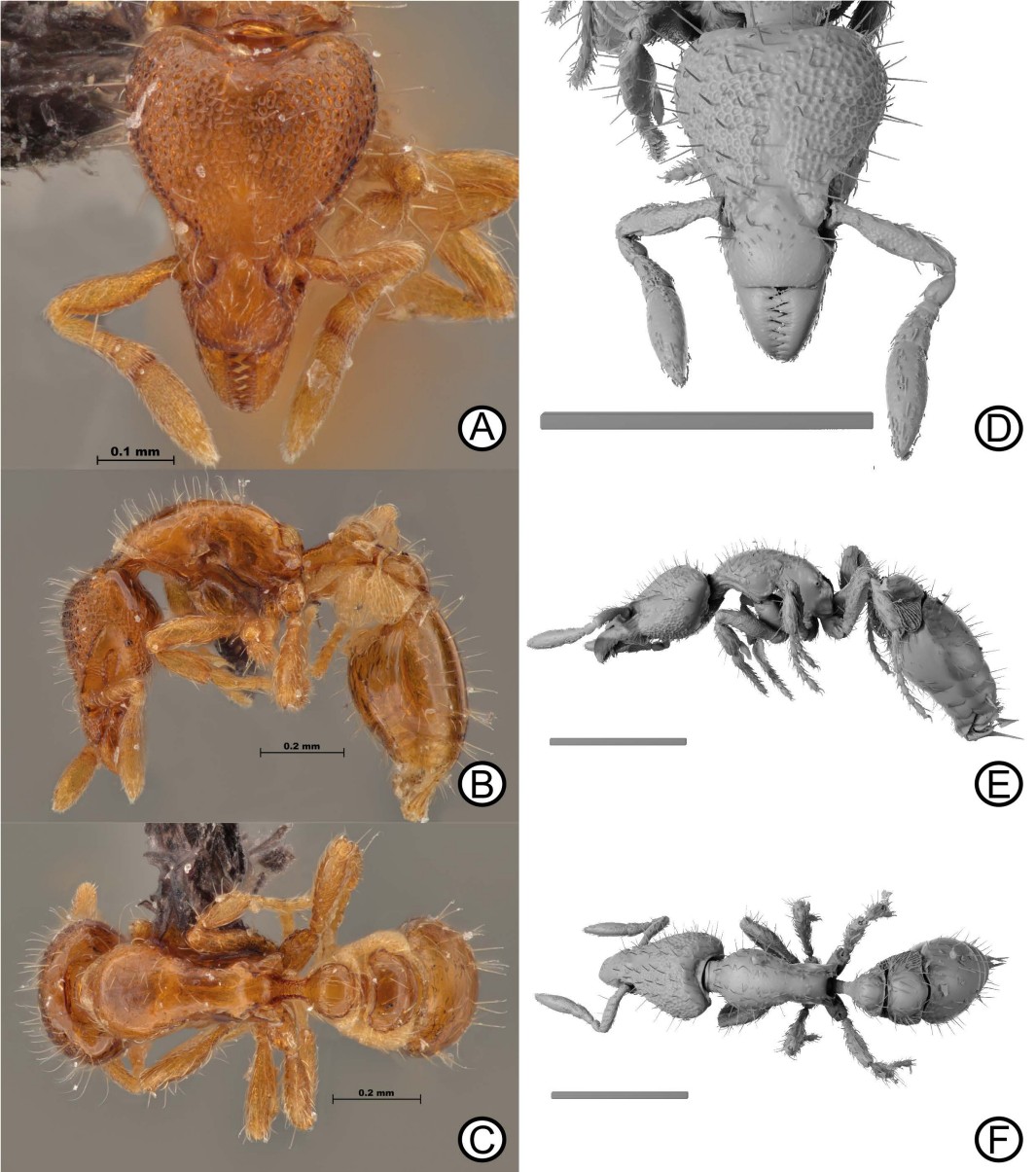

**Fig 6. *Strumigenys mazu* worker specimen images and digital micro-CT scan stills.** Digital images of *Strumigenys mazu* specimen (ANT-WEB1010081) (A) head, (B) lateral, (C) dorsal. Micro-CT scan stills of *S. mazu* specimen (CASENT0745759) (D) head, (E) lateral, (F) dorsal. Scale bars = 0.5 mm (D-F).

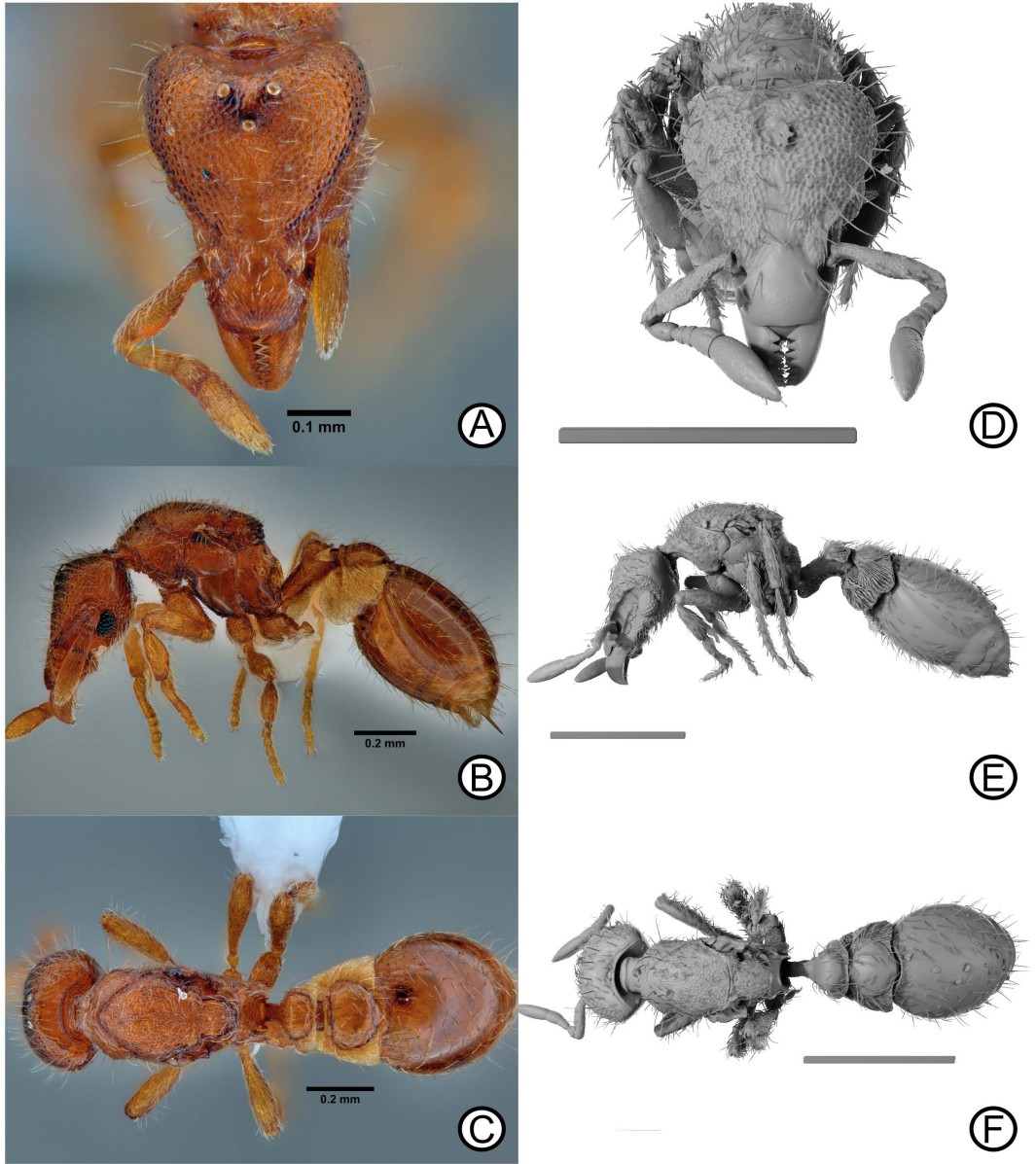

**Fig 7. *Strumigenys mazu* queen specimen (CASENT0745765).** Digital images (A) head, (B) lateral, (C) dorsal (CASENT0745765). Micro-CT scan stills (CASENT0745761) (D) head, (E) lateral, (F) dorsal. Scale bars = 0.5 mm (D-F).

## Diagnostic characters

*Strumigenys mazu* can be differentiated in the worker caste from its congeners by the reticulate-punctate sculpture on the cephalic dorsum and the absence of basigastral carinae [3,25]. Additional characters include a truncated anterior clypeal margin and a well-defined upper scrobe margin [3].

## Material examined

Specimen images examined (*n* = 1):

**Holotype** worker: Taiwan, Chilan, Yilan Hsien; 28.vii.1988; T2-F0-T4

Physical specimens examined (*n* = 17):

1 worker: China; Hong Kong, Tai Po Kau; 02 November 1979; Coll. R.Winney; soil Berlese; NHMUK010639085 [NHMUK] • 1 worker: China; Guangxi, Diding; 1110 m a.s.l; 9 July 1999; Coll. J.F.Fellowes; JF0006 [HKBM] • 1 worker: China; Hong Kong, Tai Po Kau; 22.42007 114.1829; 290 m a.s.l; 2 July 2015; Coll. T.Tsang; Winkler ex. Leaf litter; ANTWEB1010081 [IBBL] • 1 worker: China; Hong Kong, Penny's Bay; 22.31339101 114.036877; 5 m a.s.l; 25 October 2017; Coll. M.Pierce; Mini Winkler, ex leaf litter; WC-PB-INV-03 [IBBL] • 1 worker: China; Hong Kong, Kau Tam Tso; 22.5034 114.2459; 90 m a.s.l; 26 April 2022 Coll. A.I.Weemaels; Winkler ex. 0.5m² leaf litter; CASENT0764985 [IBBL] • 1 worker: China; Hong Kong, Tai Mo Shan; 22.42341 114.12825; 570 m a.s.l; 4 September 2022; Coll. M.T.Hamer; Winkler ex. random leaf litter; TMS1GC2−9 [IBBL] • 1 worker: China; Hong Kong, Tai Mo Shan; 22.42351 114.1281; 590 m a.s.l; 14 September 2022 Coll. M.T.Hamer & A.I.Weemaels; Winkler ex. 0.5m² leaf litter; TMS2T3W3-5 [IBBL]• 1 worker: China; Hong Kong, Tai Mo Shan; 22.42351 114.1281; 590 m a.s.l; 14 September 2022; Coll. M.T.Hamer & A.I.Weemaels; Winkler ex. 0.5m² leaf litter; TMS2T3W1-8 [IBBL] • 1 worker: China; Hong Kong, Tai Mo Shan; 22.42395 114.1292; 530 m a.s.l; 14 September 2022; Coll. M.T.Hamer & A.I.Weemaels; Winkler ex. 0.5m² leaf litter; TMS2T1W3-2 [IBBL] • 6 workers: China; Hong Kong, Tai Mo Shan; 22.42351 114.1281; 590 m a.s.l; 14 September 2022 Coll. M.T.Hamer; Hypogaeic Winkler, 1m² soil; ANTWEB1010145, ANTWEB1010147, ANTWEB1010148, ANTWEB1010149, ANTWEB1010150, ANTWEB1010151 [IBBL] • 1 worker: China; Hong Kong, Tai Mo Shan; 22.42351 114.1281; 590 m a.s.l; 14 September 2022 Coll. M.T.Hamer; Hypogaeic Winkler, 1m² soil; CASENT0745759 [ZRC] • 1 worker: China; Fujian, Quanzhou; 25.64504491 118.202877; 945 m a.s.l; 22 June 2023 Coll. L.Xuan; Winkler ex. 4 m² leaf litter; ANTWEB1010144 [IBBL] • 1 worker: China; Fujian, Quanzhou; 25.63946617 118.2267705; 1070 m a.s.l; 25 June 2023 Coll. L.Xuan; Winkler ex. 4 m² leaf litter; ANTWEB1010143 [IBBL].

## Virtual dataset

Worker based virtual datasets contains surface volume rendering image stacks of PNG files for CASENT0745759, comprising stills of the head in full-face view, the mandibles in anteroventral view, the whole body in lateral and dorsal view, as well as individual images of both petiolar segments, and gaster both in dorsal view. Further, raw volumetric data (nii format), and a full 3D surface model (STL format) are made available for CASENT0745759, as well as CASENT0764985. All datasets are deposited on Zenodo (https://zenodo.org/records/14830376). 3D surface models of the whole body of CASENT0745759, and CASENT0764985 are also made available on Sketchfab (CASENT0745759; https://skfb.ly/pyFxI & CASENT0764985; https://skfb.ly/pyFxK).

Worker specimens (*n* = 10): HW 0.31–0.34; HL 0.42–0.44; ML 0.07–0.09; SL 0.17–0.20; WL 0.42–0.46; PW 0.20–0.23; PL 0.20–0.29; PH 0.10–0.12; DPW 0.11–0.13; DPL 0.06–0.10; DPPW 0.17–0.19; DPPL 0.11–0.13; ATL 0.30–0.34; TL 1.54–1.70

Indices: CI 71.76–79.30; MI 16.55–20.67; SI 51.84–57.48; PI 62.35–68.37; LPI 37.41–55.67; DPI 39.16–61.62; DPPI 138.21–161.82.

## Female description

### Material examined

Physical specimens examined (*n* = 3):

1 alate gyne: CN-HK Tai Po Kau 22.4204 114.17577 295m 20−27.viii.2022 FIT Vane MTH ANTWEB1010079 [IBBL] • 1 dealate gyne: China; Hong Kong, Tai Mo Shan; 22.42351 114.1281; 590 m a.s.l; 14 September 2022 Coll. M.T.Hamer; Hypogaeic Winkler, 1m² soil ANTWEB1010146 [IBBL] • 1 dealate gyne: CN-HK Tai Mo Shan 22.42335 114.1282 598m 14.ix.2022 Winkler MTH/AIW. CASENT0745765 [ZRC] • 1 dealate gyne: CN-HK Tai Mo Shan 22.42335 114.1282 598m 14.ix.2022 Winkler MTH/AIW CASENT0745761 [IBBL] • 1 dealate gyne: CN-HK Tai Mo Shan 22.42335 114.1282 598m 14.ix.2022 Winkler MTH/AIW TMS2T3W1-9 ANTWEB1010080 [IBBL].

## Virtual dataset

*Strumigenys mazu* gyne (CASENT0745761) cyber-material datasets consist of surface volume rendering image stacks of PNG files, comprising full-face views stills of the head in full-face view, as well as whole body in lateral and dorsal view. Further, raw volumetric data (nii format), and a full 3D surface model (STL format) are made available. All data is deposited on Zenodo (https://zenodo.org/records/14830376). A 3D surface model of CASENT0745761 is also made available on Sketchfab (https://skfb.ly/pyFxL).

## Measurements

Gyne specimens (*n* = 3): HW 0.35–0.37; HL 0.44–0.47; ML 0.09–0.10; SL 0.18–0.19; WL 0.53–0.58; PW 0.26–0.27; PL 0.25–0.27; PH 0.11-0.13; DPW 0.14; DPL 0.07; DPPW 0.21–0.22; DPPL 0.13–0.14; ATL 0.42–0.45; TL 1.86–1.97. Indices: CI 75.91–82.25; MI 18.92–23.3; SI 51.27–51.91; PI 72.68–75.92; LPI 43.53–53.04; DPI 55.69–56.28; DPPI 155.8–160.15.

## Head

In full-face view, head longer than broad, broadest just anterior to posterior border. Mandibles triangular, reducing in width apically; basal masticatory margin with five stout triangular teeth, apical half with two smaller stout triangular teeth, followed by four smaller teeth and final acute apical tooth. Anterior clypeal border straight; posterior clypeal border narrowly inserted between antennal foramen. Antennae with six segments, including five funicular segments; scapes short, expanding apically; funiculus terminating in two-segment club, apical most segment conspicuously longer than broad. In lateral view, upper scrobe margin conspicuous and well defined, slightly overhanging lateral head surface. Eye present, comprising 25–30 ommatidia, located posteroventrally on lateral head surface. In lateral view scrobe present, as long as antennal scape. Preocular carina present, terminating just anterior of eye. Preocular cuticular strip conspicuous, running parallel to scrobe and terminating before median eye level. Postbuccal impression shallow, inconspicuous. Three ocelli present, distance between ocelli greater than ocelli diameter. Posterior border concave, with lamella extension within concavity directly anterior to head-pronotal neck articulation.

## Mesosoma

In dorsal view, pronotum wider than long; anterior margin convex, slightly elevated from pronotal collar, posterior margin concave. Pronotum extending to lateral surface without dorsolateral margin. In dorsal view, anterior margin of mesoscutum convex and slightly elevated from pronotal surface; posterior margin straight, widest just in front of tegula; posterior lateral margins of mesoscutum converging towards mesoscutellum. Mesoscutum-mesoscutellum sulcus wide, weakly impressed. In dorsal view, mesoscutellum just longer than wide, widest point just anterior of middle; posterior margin convex. In posterodorsal view, metanotum visible as thin strip located below mesoscutellum and above the propodeum, terminating before tegula. Propodeum longer than wide, narrow; carinate margin present between metanotum and propodeum. One pair of spongiform tissue lobes present on metapleuron-propodeal margin. In lateral view, mesopleuron divided into anepisternum and katerpisternum by narrow sulcus. Katepistermum with sharp, anteriorly directed cuticular spur almost reaching posteroventral corner of pronotal side. Narrow sulcus separating katerpisternum and metapleuron. Metapleuron undivided, rectangular in shape. Metapleural lobe small; metapleural gland bulla small, oval shape, positioned and directed anterodorsally. Spiracle positioned high on upper metapleuron.

## Metasoma

In lateral view, petiolar peduncle short, node subclubate; dorsolateral margins with carinae, not smoothly rounded. Petiolar spiracle located just below node on posterolateral region of peduncle. In dorsal view, petiolar disc wider than long; postpetiolar disc wider than long, lateral margins converging on posterior margin in gradual arc. In lateral view, spongiform tissue massively produced around petiole and postpetiole; small spongiform tissue curtain present on ventral face of

petiole. First gastral tergite long, as long as petiole and postpetiole combined; in dorsal view, anterolateral corners of first gastral tergite blunt.

## Pilosity

Clypeus with short, simple decumbent and appressed setae directed anteriorly. Scapes with decumbent and appressed setae, alongside several long, erect, simple setae on lead edge of scape. Funicular segments with dense, short pubescence. Head dorsum with long, erect, simple setae slightly curved at apex; shorter, erect, simple setae also present among longer setae. In full-face view, eight long, erect simple setae located along upper scrobe margin arranged serially and directed anterolaterally. Corners of posterior head margin with medium length, subdecumbent and fully appressed simple setae, noticeably different to surrounded erect setae. Mesosomal dorsum covered in long, erect, simple setae, plus shorter, erect simple setae and semi-decumbent short, simple setae among erect, longer setae. Pronotal humeral seta long, erect, longer than surrounding setae and directed anterolaterally. Lateral mesosomal surface almost seta-less, some setae present on dorsolateral margin zone; short, simple setae present just above propodeal spiracle; katerpisternum and anasepsiternum entirely glabrate. Short, subdecumbant setae on anterior petiole face; petiole and postpetiole dorsum with long, simple setae. Gastral dorsum with long, erect simple setae, noticeably longer than those on mesosomal and head dorsa; gaster ventrally with medium length appressed setae and erect medium length setae. Coxa, trochanter, femur and tibia with numerous, dense and appressed long setae; several conspicuous long, projecting setae also present among the appressed setae.

## Sculpture

Mandibular dorsum with dense micro-reticulation and non-reflective. Clypeus smooth centrally, with micro-punctures on periphery. Cephalic dorsum from posterior clypeal border to posterior lobes reticulate-punctuate; small region immediately behind posterior clypeal border smooth. Scapes and funicular segments with dense microsculpture. Pronotum partially smooth, with some microsculpture on lateral sides. Mesoscutum and mesoscutellum reticulate-punctuate both dorsally and laterally. Katerpisternum, anespisternum and metapleuron smooth, partial microsculpture above metapleural gland bulla. Propodeal surface smooth. Petiolar peduncle laterally punctuate in the upper and smooth in the lower half; anterior face dorsally smooth. Petiole and postpetiole dorsally smooth. Gaster dorsally smooth; basigastral costulae present, short, hardly extending 1/5th of the first gastral tergite.

## Colour

Body and appendages reddish orange, spongiform tissue light yellow.

## Distribution

Known from China (Hong Kong SAR, Guangxi Province, Fujian Province [New record]); Taiwan; Japan; Vietnam (Fig 1; Table 1).

## Comments

*Strumigenys mazu* is the most widespread species within this species group, known from Japan, Mainland China (Fujian, Hong Kong SAR, Guangxi), Taiwan and Vietnam. Specimens were collected within leaf litter and soil, from broadleaf forest floors in Japan [24], and similarly in Hong Kong [22,26]. Collection localities include high elevation sites, e.g., Tai Mo Shan (560 m a.s.l), as well as some lowland sites including Kau Tam Tso (93 m a.s.l) and Penny's Bay (10 m a.s.l). Masuko [24] noted, on the basis of a single colony, that those were small, comprising a single queen and 19 workers, whereas the maximum number of individuals of *S. mazu* collected in one soil samples (1 m²) from

Hong Kong was 44 and one dealate gyne. It is likely that this sample represented a single colony, especially given their rarity across Hong Kong. This species likely feeds on mites with Masuko [24] finding gamasid mites in field examinations of *S. mazu* nests, but under lab conditions also accepted collembolans and symphylans. A description of the gyne (Fig 7A-F) is provided here. The first record of *S. mazu* from Fujian Province, China is also provided, where it was collected in two localities in Daiyunshan Nature Reserve within broadleaf forests at relatively high elevations (943–1074 m a.s.l). The occurrence of *S. mazu* in Fujian confirms the prediction of Tang & Guénard [26]. No records are yet known from Guangdong Province, but it seems very likely that this species should be present given its occurrence in Guangxi, Fujian, and Hong Kong.

**Strumigenys mnemosyne** (Bolton, 2000)

*Pyramica mnemosyne* Bolton, 2000: 446, figs. 266, 291 (w.) BORNEO (East Malaysia: Sarawak).
Combination in *Strumigenys*: Baroni Urbani & De Andrade, 2007: 124 Fig 2D–F.

### Diagnostic characters

*Strumigenys mnemosyne* is morphologically close to *S. marmorata,* both lack standing setae on the scape, sculpture on the head and mesosomal dorsum. Each species can be differentiated from one another by the presence of erect simple setae on the cephalic, mesosomal and petiole dorsum in *S. mnemosyne* and the conspicuously wider head in *S. marmorata* (HW 0.34–0.39). Additional differing characters include the shape of the scape, which is heavily constricted proximally in *S. mnemosyne*, the significantly longer setae on the posterior of the first gastral tergite and subsequent tergites.

### Material examined

Physical specimens examined (*n* = 3):

**Holotype** worker: Malaysia; Sarawak, 4th Division, Gn. Mulu Nat. Pk, RGS Expd., Long Pala, 26.ix.1977, lowland rainforest leaf litter (B. Bolton) CASENT0900146 [NHMUK] • 1 worker: Malaysia; Sabah, Deramakot Forest Reserve, January – April 1998; Brühl; NHMUK010639089 [NHMUK] • 1 worker: Malaysia; Sabah, Poring Hot Spring, 10/5/1987; Burokhardt & Löbl; NHMUK010639088 [NHMUK].

### Distribution

Known only from East Malaysia (Sabah & Sarawak) (Fig 1; Table 1).

### Comments

As far as the authors are aware, *S. mnemosyne* is only known from Sabah and Sarawak in Borneo from four localities, Long Pala, Maliau Basin (CASENT0394454 & CASENT0700789, AntWeb.org), Deramakot Forest Reserve and Poring Hot Spring. Like other members of the group, little is known of the biology of *S. mnemosyne* but all specimens with collection information are known from leaf litter extractions.

**Strumigenys rimdahli sp. nov.** (Hamer, Tang & Guénard, 2025)

urn:lsid:zoobank.org:act:619FD6BE-56A9-48C7-8622-DF68FA7D6D72 Fig 8A–F.

### Diagnosis

In full face-view, head longer than broad, broadest on posterior third. Scape lacking standing setae. Cephalic dorsum with sparse standing setae; in full-face view, dorsolateral margin with single pair of projecting setae that extend beyond dorsolateral margin. Dorsolateral margin of mesosoma distinctly marginated; in lateral view, mesosomal dorsum flat.

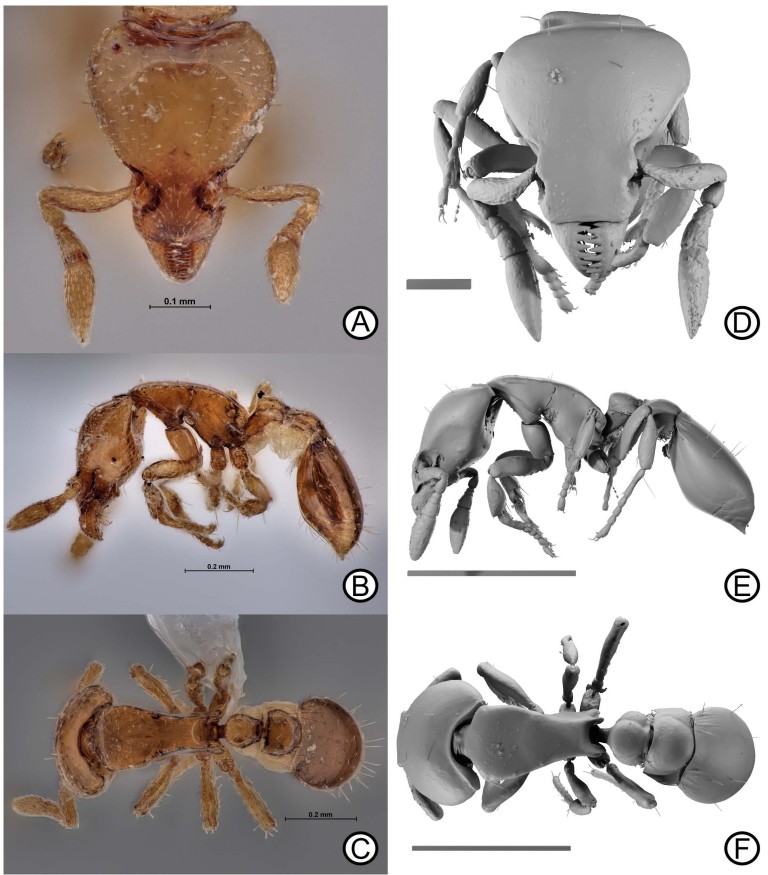

**Fig 8. Holotype specimen (ANTWEB1010648) of *Strumigenys rimdahli* sp. nov.** Digital images (A) head, (B) lateral, (C) dorsal. Micro-CT scan stills (D) head, (E) lateral, (F) dorsal. Scale bars = 0.1 mm (D), 0.5 mm (E, F).

Anterolateral margin of pronotum angulated in dorsal view. Anterior margin of postpetiolar disc conspicuously straight anteriorly. Basigastral costulae present. Dorsal surface of meso- and metabasitarsus with single, long erect seta. Meso- and metatibia with single erect seta noticeably shorter than erect seta on basitarsus. Legs with numerous appressed, decumbent and semierect setae on all segments.

## Material examined

1 **Holotype** worker: TH; Chiang Mai; N19.236638 E98.650739; ±20m²; 1106 masl; 10.viii.2024; LLex.4m2; MTHamer et al; old sec. for. CM-P17-01. ANTWEB1010648 [ZRC].

## Cybertypes

Holotype (ANTWEB1010648) dataset consists of surface volume rendering image stacks of PNG files comprising full-face views stills of the head, whole body in lateral and dorsal view, as well as mandibles in anterior view. Further, raw volumetric data (nii format), and a full 3D surface model (STL format) are made available. Digital stacked colour images of the whole body in lateral and dorsal view, as well as the head in full-face view are also provided (TIFF format). All data is deposited on Zenodo (https://zenodo.org/records/14830376). A 3D surface model is also made available on Sketchfab (https://skfb.ly/pyFxM).

## Description

Holotype measurements: HW 0.32; HL 0.37; ML 0.06; SL 0.16; WL 0.37; PW 0.18; PL 0.17; PH 0.10; DPW 0.09; DPL 0.09; DPPW 0.13; DPPL 0.09; ATL 0.30; TL 1.36.

   Indices; CI 84.68; MI 16.67; SI 50.16; PI 57.46; LPI 58.43; DPI 56.02; DPPI 142.20.

## Head

In full face-view, head longer than broad, broadest on posterior third. Posterior margin broadly concave, with region of transparent cuticle. Lateral clypeal margins converging anteriorly; anterior margin subtly convex. Epistomal sulcus absent, cephalic dorsum and clypeus continuous; torular lobe small, convex and obscuring antennal foramen in full-face view. Antennae totalling six segments; scape short, failing to reach posterior margin; scape dorsoventrally flattened, converging anteriorly to form flange at leading edge, narrowing, basally and distally. Flagellomeres terminating in two-segment club; apicalmost segment longer than penultimate segment. Mandible triangular, reducing in width apically; basal masticatory margin with one short triangular basal tooth and four long, triangular teeth, apical half with two triangular teeth of equal length to previous four teeth, followed by four small teeth and an apical tooth the same size as previous two teeth. In lateral view, preocular carina short, terminating long before eye level. Glandular groove not well impressed, terminating well before eye. Eye minute, comprised of single ommatidium. Postbuccal impression conspicuous; ventral head margin, from preocular notch to occipital foramen sinuate, being convex before eye and concave anterior to occipital foramen. In lateral view, angle of intercept between ventrolateral head margin and posterior head margin slightly obtuse, with a minute posterior projecting cuticular lobe at intercept.

## Mesosoma

In dorsal view, mesosoma lacking promesonotal suture, metanotal groove absent. In lateral view, promesonotal suture weakly impressed, not reaching dorsolateral margin. Mesosomal outline weakly convex anteriorly to strongly convex posteriorly due to propodeal declivity. Dorsolateral margin distinctly marginated. In lateral view, mesosomal dorsum flat. Mesopleural notch present and well impressed; seta wheel present. Katepistermum with a long, sharp, anteriorly directed cuticular spur almost reaching posteroventral corner of pronotal side. In dorsal view, pronotum reaching maximum width at mid-length; angulated anteriorly with anterolateral angle obtuse. Propodeum unarmed; propodeum with semi-rectangular, dorsally rounded spongiform tissue only. Propodeal declivity slightly concave in dorsal view.

## Metasoma

In lateral view, petiolar peduncle and anterior petiolar face meeting at obtuse angle; anterior face of petiole flat, grading into dorsal face with blunt angle. Dorsal margin of petiolar and postpetiolar surfaces convex. In dorsal view, petiolar disc oval, wider than long; postpetiolar disc wider than long, anterior margin straight, with weakly posteriorly converging lateral margins, posterior margin feebly concave. Ventral petiolar spongiform tissue present; lateral petiole spongiform tissue small. Conspicuous ventral and lateral spongiform tissue on postpetiole, obscuring lateral postpetiolar surface. Spongiform tissue surrounding postpetiolar disc both anteriorly and laterally in dorsal view, but not on top of postpetiole disc itself.

## Pilosity

In lateral view, cephalic dorsum with two pairs of standing setae. In full-face view, dorsolateral margin with single standing seta; cephalic dorsum with short and appressed setae, concentrated posteriorly on cephalic dorsum, directed inwards to cephalic mid-line. Clypeal dorsum lacking standing setae, setae instead simple, appressed setae present, directed anteriorly. Setae on both cephalic and clypeal dorsa well-spaced. Ventral cephalic setae mix of short, appressed setae alongside long, appressed simple setae. Postbuccal cavity with conspicuous series of long and erect simple setae noticeably longer than surrounding setae on cephalic dorsum. Scape lacking standing setae, appressed and decumbent, simple setae

present only. Flagellomere segments with short, appressed simple setae, reaching highest concentration on apical segment. Pronotal humeral seta present, simple and apically blunt, directed antero-laterally and well differentiated from surrounding appressed mesosomal setae. Dorsal mesosomal setae appressed and simple, directed inwards to mesosomal longitudinal midline. Hair wheel with four to five, appressed simple setae. Petiole and postpetiole disc with sparse, simple erect setae; single pair of setae located on posterior margin of postpetiole dorsum directed posterodorsally. Gastral tergites with long, stout erect setae and short appressed setae. Gastral sternites in lateral view, with long, erect simple setae, noticeably shorter and less stout than those on tergites. Gastral sternites also with long, decumbent to semi-erect simple setae. Meso- and metabasitarsus with single, long projecting seta. Meso- and metatibia with single erect seta noticeably shorter than the projecting seta on basitarsus. Legs also covered in numerous appressed, decumbent and semierect setae on all segments.

## Colour

Body colour reddish brown, appendages light reddish brown, spongiform tissue whitish yellow.

## Sculpture

Almost entire body integument lacking sculpture, only sculpture located on first gastral tergite in form of basigastral costulae, and micro-punctures on all antennal segments, coxae and basitarsi.

## Etymology

Named after Mr. Kenneth Rimdahl, founder of Monsoon Tea, in recognition of his conservation efforts and help in facilitating fieldwork in Thailand.

## Distribution

Only known from Chiang Mai, Thailand (Fig 1; Table 1).

## Comments

*Strumigenys rimdahli* is a member of the *mnemosyne* group, displaying the necessary morphological characters to be placed into the group. Within the group, *S. rimdahli* is most similar to S. *mnemosyne* and *S. marmorata* but can be easily differentiated from both by the conspicuous marginated dorsolateral mesosomal margin, and the flat mesosomal dorsum. Further characters that differentiate *S. rimdahli* from S. *mnemosyne* include the single, stout seta extending beyond lateral cephalic margin (four setae in S. *mnemosyne*), alongside the reduced length and abundance of standing setae across the whole body most especially the legs and mesosoma dorsum (including the pronotal humeral setae). In *S. rimdahli* the anterior corners of the pronotal dorsal margin are angulated whereas in *S. mnemosyne* they are more rounded. Morphological characters that differentiate *S. rimdahli* from *S. marmorata* include the presence of long, freely projecting setae on the outer surfaces of the meso- and meta- tibae and basitarsi (absent in *S. marmorata*), the greater number of standing setae on the head dorsum and legs, and a straight anterior petiole margin in dorsal view (concave in *S. marmorata*).

## Ecology

The holotype specimen was collected in an old secondary forest in Mae Sae, Chiang Mai Province at 1106 m a.s.l within a 4 m² leaf litter sample extracted within a Winkler over 72 hours. Nothing else is known about the biology of *S. rimdahli*.

**Strumigenys runa** (Bolton, 2000)

*Pyramica runa* Bolton, 2000: 447 (w.) BORNEO (East Malaysia: Sabah).
Combination in *Strumigenys*: Baroni Urbani & De Andrade, 2007: 127 Fig 9A–C.

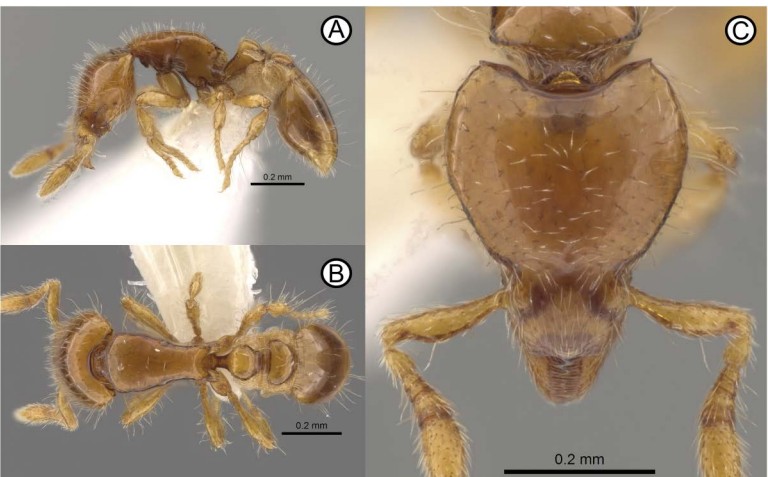

**Fig 9. *Strumigenys runa* specimen images (CASENT0634746).** Digital images (A) lateral view, (B) dorsal view, (C) full-face view. Photographer John T. Longino.

### Diagnostic characters

*Strumigenys runa* is morphologically close to *S. mnemosyne* but differs in the leading edge of the scape with 1–2 elongate straight setae that project anterodorsally. Pronotal dorsum shallowly concave in lateral view. Lack of distinct, paired setae on mesosomal dorsum. Cephalic setae fine, longer and standing, much denser than in *S. mnemosyne* and not arranged in rows [3].

### Material examined

Specimen images examined (*n* = 1):

   1 worker: Malaysia: Sabah: Nepenthes Camp, Maliau Basin; 4.73356 116.87757 ± 50m 1010m (a.s.l); 27 July 2014; J. Longino #8798-s. Wet Forest, ex sifted leaf litter; CASENT0634746 [JTLC]

### Distribution

Only known from East Malaysia (Sabah) (Fig 1; Table 1).

### Comments

*Strumigenys runa* is only known from the island of Sabah in the north of the island of Borneo. Of the known specimens, two are from a collection site above 1000 m a.s.l. and collected within a Winkler sample. Little more is known about the biology of *S. runa*. In this study, the first high resolution images of *S. runa* are provided, supplied by Professor John T. Longino.

***Strumigenys taphra*** (Bolton, 2000)
   *Pyramica taphra* Bolton, 2000: 448 (w.) Thailand. Fig 2G–I.

### Diagnostic characters

In terms of total size, *S. taphra* is the largest of the *S. mnemosyne* group. Other distinguishable characters include the standing setae on the clypeus and inconspicuous punctate sculpturing on the cephalic dorsum just posterior to clypeus. Cephalic sculpture is present in *S. mazu* but it is significantly more conspicuous and distributed across the head dorsum.

*Strumigenys taphra* can easily be separated from *S. marmorata* by the deeply concave posterior head margin and the presence of a long, glandular groove which extends beyond the posterior margin of the eye as well as its larger total size.

## Material examined

Specimen images examined (*n* = 1):

**Holotype** worker: Thailand; Chiang Mai Prov., Chom Thong, Doi Inthanon, 12.vii.1991, CM 141, 1700 m., soil berl., rainfor., Deharveng & Bedos CASENT0915365 [MNHN]

Physical specimens examined (*n* = 8):

2 **Paratype** workers from Thailand: Chiang Mai Prov., Chom Thong, Doi Inthanon, 12.vii.1991, CM 141, 1700 m., soil berl., rainfor., Deharveng & Bedos; CASENT0900147 [NHMUK] • 3 **Paratype** workers: Thailand: Chiang Mai Prov., Chom Thong, Doi Inthanon, 12.vii.1991, CM 141, 1700 m., soil berl., rainfor., Deharveng & Bedos NHMUK010639086 [NHMUK] • 3 **Paratype** workers from Thailand: Chiang Mai Prov., Chom Thong, Doi Inthanon, 12.vii.1991, CM 141, 1700 m., soil berl., rainfor., Deharveng & Bedos NHMUK010639087 [NHMUK].

## Distribution

Only known from Chiang Mai, Thailand (Fig 1; Table 1)

## Comments

*Strumigenys taphra* is so far only known from Chiang Mai Province, Thailand. An additional specimen (MCZ-ENT00759758) previously identified to *S. taphra* known from Yunnan, China is here reassessed to be a new species (see *S. liui* description). Like other members of the *S. mnemosyne* group, *S. taphra* was collected from soil, and is thus likely to be a species that occupies soil and leaf litter. Unfortunately, no new specimens were found on a recent trip to Chiang Mai Province.

## Key to species of the *Strumigenys mnemosyne* group based on worker caste

(Adapted from the Southeast Asia *Strumigenys* key of Bolton [3] branching from couplet 84 and ending at couplet 87)

1. Cephalic dorsum behind clypeus uniformly finely and densely reticulate-punctate (Fig 6A, D); pronotum smooth and shining, contrasting strongly with the head (Fig 6B)...... ***Strumigenys mazu***

   • Cephalic dorsum behind clypeus and pronotum smooth and shining, the two not contrasting (Fig 2A–I; Fig 3A–C; Fig 4A–C; Fig 8A–C; Fig 9A–C)...... **2**

2. Leading edge of antennal scape without straight standing setae of any form (Fig 2D; Fig 4A; Fig 8A)...... **3**

   • Leading edge of antennal scape with 2–3 straight standing setae that freely project anterodorsally Fig 2A, G; Fig 3A, D; Fig 9C).... **5**

3. Dorsolateral margin of mesosoma marginated (Fig 8B–C, E–F); in lateral view, mesosomal dorsum flat (Fig 8B, E)… ***Strumigenys rimdahli*** sp. nov.

   • Dorsolateral margin of mesosoma curved (Fig 2E; Fig 4B); in lateral view, mesosomal dorsum convex (Fig 2E; Fig 4B); … **4**

4. In full-face view, head with four freely projecting setae that extend beyond the lateral cephalic margin (Fig 2D); HW 0.30… ***Strumigenys mnemosyne***

   • In full-face view, head without freely projecting setae that extend beyond the lateral cephalic margin (Fig 4A, F); HW 0.34–0.39 ……… ***Strumigenys marmorata*** sp. nov

5. Distinctly larger species with a narrow head and longer scape, HL 0.48–0.52; CI 70–74; SI 59–63 …. **6**

   • Smaller species with relatively broader head and shorter scape; HL 0.40–0.42; CI 80–81; SI 50–53 …**7**

6. Dorsum of clypeus in profile with short fine erect setae present (Fig 2H). In full-face view, lateral margin of clypeus with 1–2 straight setae that project out beyond the margin (Fig 2G). Basigastral costulae present (Fig 2I)..... ***Strumigenys taphra***

   • Dorsum of clypeus in profile without short fine erect setae, appressed setae only (Fig 3B). In full-face view, lateral margin of clypeus without 1-2 straight setae that project beyond the margin (Fig 3A, D). Basigastral costulae absent (Fig 3C, F) …. ***Strumigenys liui*** sp. nov.

7. Posterior margin of head on each side of the median concavity with a translucent blister-like area that occupies about half the width of the posterior margin of the posterior lobes (Fig 2A) …….. ***Strumigenys daspleta***

   • Posterior margin of head on each side of the median concavity with a narrow marginal carina across the whole width of the posterior margin of the posterior lobes (Fig 9C)...... ***Strumigenys runa***

## Discussion

The *Strumigenys mnemosyne* group is likely the most inconspicuous and one with the smallest body size among *Strumigenys* species groups in southeast Asia with all workers smaller than 2 mm. Eight species are now known from the group, of which three new species are described within this study. All non-winged individuals were collected in leaf litter and soil samples. Despite extensive leaf litter collecting and extraction campaigns across Hong Kong and Macau comprising thousands of samples, relatively few *S. mazu* and only two specimens of *S. marmorata* were collected. The use of few hypogaeic Winkler samples (comprising 1 × 1 × 0.05 m of soil, excluding leaf litter) did however provide near immediate records of *S. mazu.* Other members of the group, e.g., *S. taphra*, are also known from soil samples. The *mnemosyne* group also exhibit various morphological characteristic that are more associated with subterranean life, including shortened, triangular mandibles, highly reduced eyes, small size, and the absence of propodeal spines [27]. Considering collections in both soil and leaf litter of which the first few layers of soil are often collected), as well as a suite of morphological characters associated with subterranean life, the *S. mnemosyne* group are very likely to be hypogaeic. The deployment over 500 baited subterranean traps across Hong Kong and Macau has however yielded no members of the *S. mnemosyne* group [28], Guénard, unpublished data]. It could be that members of the group are scarce, and/or low in ant competition hierarchies with subterranean baited traps often catching more competitive species (e.g., *Carebara* spp. [former *Pheidologeton*] and *Pheidole* spp.); or that the baits used were not responding to the specialized diet on live micro-arthropods that these species may present. Their rarity has undeniably contributed to lack of genetic information of the group, leading to their phylogenetic position within *Strumigenys* currently unknown [6]. Increasing the use of soil extraction procedures across Southeast Asia will undoubtedly reveal more new species and new records for this group.

   Our micro-CT scanning approach was successful at revealing minute morphological characters (< 0.1 mm) such as the mandibular configuration of *S. marmorata*. This enabled a more comprehensive description of the new species and the slight expansion of the species group definition to encompass subtle dentition variation in *S. marmorata*. Further, a broken paratype specimen was successfully repaired in 3D virtual space using data derived from micro-CT (Fig 5). Virtual repair of important specimens has, as far as the authors are aware, not been explored prior to this study, and increases the value of micro-CT scanning as a tool for systematists where valuable specimens (such as types) are physically broken. Such virtual repairing workflows could be applied in the future where broken type material, particularly historical type specimens, might be present.

## Acknowledgments

The first author would like to thank Kenneth Rimdahl and colleagues at Monsoon Tea for facilitating fieldwork in Thailand, as well as Kelsey Davies and Marco Chan for their invaluable assistance during fieldwork. The authors are also thankful to Professor John T. Longino for specimen images of *Strumigenys runa*. We are also thankful to Crystal Maier from the Museum of Comparative Zoology, Harvard University for sending us specimens of *S. liui*. The first author is also grateful to Suzanne Ryder for allowing access and facilitating the examination of the *Strumigenys* NHMUK collection. All authors would like to thank all those who collected specimens throughout Southeast Asia used within this study. Furthermore, we thank the Imaging Section of the Okinawa Institute of Science and Technology Graduate University (OIST) for providing access to the Zeiss Xradia micro-CT scanner used for this study, and in particular Shinya Komoto for general support.

## Author contributions

**Conceptualization:** Matthew T. Hamer, Kit Lam Tang, Francisco Hita-Garcia, Benoit Guénard.

**Data curation:** Matthew T. Hamer, Julian Katzke.

**Formal analysis:** Matthew T. Hamer, Julian Katzke.

**Funding acquisition:** Benoit Guénard.

**Investigation:** Matthew T. Hamer, Julian Katzke, André Ibáñez Weemaels.

**Methodology:** Matthew T. Hamer, Julian Katzke, Benoit Guénard.

**Resources:** Evan P. Economo.

**Supervision:** Francisco Hita-Garcia, Evan P. Economo, Benoit Guénard.

**Validation:** Kit Lam Tang, Francisco Hita-Garcia, Evan P. Economo, Benoit Guénard.

**Visualization:** Matthew T. Hamer, Julian Katzke, André Ibáñez Weemaels.

**Writing – original draft:** Matthew T. Hamer, Julian Katzke.

**Writing – review & editing:** Matthew T. Hamer, Julian Katzke, Kit Lam Tang, André Ibáñez Weemaels, Francisco Hita-Garcia, Evan P. Economo, Benoit Guénard.

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
