## [Decision Letter · Decision Letter 0]

17 Jun 2025

PONE-D-25-13901

A revision of the rare Strumigenys mnemosyne (Formicidae; Myrmicinae) group using micro-CT scanning, with the description of three new species, and the virtual repair of a broken paratype.

PLOS ONE

 Dear Dr. Hamer, Thank you for submitting your manuscript to PLOS ONE. After careful consideration, we feel that it has merit but does not fully meet PLOS ONE’s publication criteria as it currently stands. Therefore, we invite you to submit a revised version of the manuscript that addresses the points raised during the review process.

**Thank you for submitting your manuscript on ‘A revision of the rare Strumigenys mnemosyne (Formicidae; Myrmicinae) group using micro-CT scanning, with the description of three new species, and the virtual repair of a broken paratype’ to PLOS ONE. After careful review by the three reviewers I came to the decision that the manuscript needs some serious revisions followed by an additional round of review. Please check the comments raised by each of the reviewers particularly the one recommended to reject the manuscript. He has raised some serious comments regarding the methodology of the manuscript. Please provide a detailed letter describing your responses to the issues that were raised by the reviewers and also please comment on any issues you did not want to incorporate into your revision. I would like to receive the revised version of the manuscript soon.**

We look forward to receiving your revised manuscript.

Kind regards,

A.P. Ranjith

Academic Editor

PLOS ONE

**Journal Requirements:**

1. When submitting your revision, we need you to address these additional requirements. Please ensure that your manuscript meets PLOS ONE's style requirements, including those for file naming. The PLOS ONE style templates can be found at https://journals.plos.org/plosone/s/file?id=wjVg/PLOSOne_formatting_sample_main_body.pdf and https://journals.plos.org/plosone/s/file?id=ba62/PLOSOne_formatting_sample_title_authors_affiliations.pdf 2. Please take this opportunity to be sure you have met all of our guidelines for new species. For proper registration of a new zoological taxon, we require two specific statements to be included in your manuscript. a) In the Results section, the globally unique identifier (GUID), currently in the form of a Life Science Identifier (LSID), should be listed under the new species name, for example: Anochetus boltoni Fisher sp. nov. urn:lsid:zoobank.org:act:B6C072CF-1CA6-40C7-8396-534E91EF7FBBAnother LSID for the manuscript itself should also appear within the Nomenclature statement. You will need to contact Zoobank (zoobank.org/About) to obtain a GUID (LSID). You should receive one LSID for your manuscript and a separate, unique LSID for the new species.  b) Please also insert the following text into the Methods section, in a sub-section to be called "Nomenclatural Acts": The electronic edition of this article conforms to the requirements of the amended International Code of Zoological Nomenclature, and hence the new names contained herein are available under that Code from the electronic edition of this article. This published work and the nomenclatural acts it contains have been registered in ZooBank, the online registration system for the ICZN. The ZooBank LSIDs (Life Science Identifiers) can be resolved and the associated information viewed through any standard web browser by appending the LSID to the prefix "http://zoobank.org/". The LSID for this publication is: urn:lsid:zoobank.org:pub: XXXXXXX. The electronic edition of this work was published in a journal with an ISSN, and has been archived and is available from the following digital repositories: PubMed Central, LOCKSS [author to insert any additional repositories]. All PLOS ONE articles are deposited in PubMed Central and LOCKSS. If your institute, or those of your co-authors, has its own repository, we recommend that you also deposit the published online article there and include the name in your article.Following a recent ruling by the International Commission on Zoological Nomenclature, electronic journals are now a valid format for publication of new zoological taxa. In order to ensure the valid publication of your new species, please be sure to include the updated version of Nomenclatural Acts (above). A complete explanation of our guidelines for publishing new species can be found on our website: http://www.plosone.org/static/guidelines#zoological. 3. Thank you for stating the following in the Acknowledgments Section of your manuscript: Environment and Conservation Fund from the Government of the Hong Kong Special Administrative Region, under the ‘Environmental Research, Technology Demonstration and Conference Projects’ funding scheme, Project number ECF 137/2020 and the Research Grant Council from the Government of the Hong Kong Special Administrative Region, GRF17103223. The first author would like to thank Kenneth Rimdahl and colleagues at Monsoon Tea for facilitating fieldwork in Thailand, as well as Kelsey Davies and Marco Chan for their invaluable assistance in the field. The authors are also thankful to Professor John T. Longino for specimen images of Strumigenys runa. We also thankful to Crystal Maier from the Museum of Comparative Zoology, Harvard University for sending us specimens of S. liui. The first author is also grateful to Suzanne Ryder for allowing access and facilitating the examination of the Strumigenys NHMUK collection. All authors would like to thank all those who collected specimens throughout Southeast Asia used within this study. Furthermore, we thank the Imaging Section of the Okinawa Institute of Science and Technology Graduate University (OIST) for providing access to the Zeiss Xradia micro-CT scanner used for this study, and in particular Shinya Komoto for general support. We note that you have provided funding information that is not currently declared in your Funding Statement. However, funding information should not appear in the Acknowledgments section or other areas of your manuscript. We will only publish funding information present in the Funding Statement section of the online submission form. Please remove any funding-related text from the manuscript and let us know how you would like to update your Funding Statement. Currently, your Funding Statement reads as follows: BG was awarded; Environment and Conservation Fund from the Government of the Hong Kong Special Administrative Region, under the ‘Environmental Research, Technology Demonstration and Conference Projects’ funding scheme, Project number ECF 137/2020. URL; https://www.ecf.gov.hk/en. Funder did not play any role in the study design, data collection and analysis, decision to publish, or preparation of the manuscript. MTH and AIW recieved salary from ECF 137/2020.BG was awarded; Research Grant Council from the Government of the Hong Kong Special Administrative Region, GRF17103223. URL https://www.ugc.edu.hk/eng/rgc/funding_opport/grf/. Funder did not play any role in the study design, data collection and analysis, decision to publish, or preparation of the manuscript. MTH recieved salary from GRF17103223. Please include your amended statements within your cover letter; we will change the online submission form on your behalf. 4. When completing the data availability statement of the submission form, you indicated that you will make your data available on acceptance. We strongly recommend all authors decide on a data sharing plan before acceptance, as the process can be lengthy and hold up publication timelines. Please note that, though access restrictions are acceptable now, your entire data will need to be made freely accessible if your manuscript is accepted for publication. This policy applies to all data except where public deposition would breach compliance with the protocol approved by your research ethics board. If you are unable to adhere to our open data policy, please kindly revise your statement to explain your reasoning and we will seek the editor's input on an exemption. Please be assured that, once you have provided your new statement, the assessment of your exemption will not hold up the peer review process. 5. We note that Figure 1 in your submission contain map images which may be copyrighted. All PLOS content is published under the Creative Commons Attribution License (CC BY 4.0), which means that the manuscript, images, and Supporting Information files will be freely available online, and any third party is permitted to access, download, copy, distribute, and use these materials in any way, even commercially, with proper attribution. For these reasons, we cannot publish previously copyrighted maps or satellite images created using proprietary data, such as Google software (Google Maps, Street View, and Earth). For more information, see our copyright guidelines: http://journals.plos.org/plosone/s/licenses-and-copyright. We require you to either present written permission from the copyright holder to publish these figures specifically under the CC BY 4.0 license, or remove the figures from your submission: a. You may seek permission from the original copyright holder of Figure 1 to publish the content specifically under the CC BY 4.0 license.   We recommend that you contact the original copyright holder with the Content Permission Form (http://journals.plos.org/plosone/s/file?id=7c09/content-permission-form.pdf) and the following text:“I request permission for the open-access journal PLOS ONE to publish XXX under the Creative Commons Attribution License (CCAL) CC BY 4.0 (http://creativecommons.org/licenses/by/4.0/). Please be aware that this license allows unrestricted use and distribution, even commercially, by third parties. Please reply and provide explicit written permission to publish XXX under a CC BY license and complete the attached form.” Please upload the completed Content Permission Form or other proof of granted permissions as an "Other" file with your submission. In the figure caption of the copyrighted figure, please include the following text: “Reprinted from [ref] under a CC BY license, with permission from [name of publisher], original copyright [original copyright year].” b. If you are unable to obtain permission from the original copyright holder to publish these figures under the CC BY 4.0 license or if the copyright holder’s requirements are incompatible with the CC BY 4.0 license, please either i) remove the figure or ii) supply a replacement figure that complies with the CC BY 4.0 license. Please check copyright information on all replacement figures and update the figure caption with source information. If applicable, please specify in the figure caption text when a figure is similar but not identical to the original image and is therefore for illustrative purposes only.The following resources for replacing copyrighted map figures may be helpful: USGS National Map Viewer (public domain): http://viewer.nationalmap.gov/viewer/The Gateway to Astronaut Photography of Earth (public domain): http://eol.jsc.nasa.gov/sseop/clickmap/Maps at the CIA (public domain): https://www.cia.gov/library/publications/the-world-factbook/index.html and https://www.cia.gov/library/publications/cia-maps-publications/index.htmlNASA Earth Observatory (public domain): http://earthobservatory.nasa.gov/Landsat:
http://landsat.visibleearth.nasa.gov/USGS EROS (Earth Resources Observatory and Science (EROS) Center) (public domain): http://eros.usgs.gov/#Natural Earth (public domain): http://www.naturalearthdata.com/ 6. We note that Figures 2 to 9 in your submission contain copyrighted images. All PLOS content is published under the Creative Commons Attribution License (CC BY 4.0), which means that the manuscript, images, and Supporting Information files will be freely available online, and any third party is permitted to access, download, copy, distribute, and use these materials in any way, even commercially, with proper attribution. For more information, see our copyright guidelines: http://journals.plos.org/plosone/s/licenses-and-copyright. We require you to either present written permission from the copyright holder to publish these figures specifically under the CC BY 4.0 license, or remove the figures from your submission: a. You may seek permission from the original copyright holder of Figures 2 to 9 to publish the content specifically under the CC BY 4.0 license.  We recommend that you contact the original copyright holder with the Content Permission Form (http://journals.plos.org/plosone/s/file?id=7c09/content-permission-form.pdf) and the following text:“I request permission for the open-access journal PLOS ONE to publish XXX under the Creative Commons Attribution License (CCAL) CC BY 4.0 (http://creativecommons.org/licenses/by/4.0/). Please be aware that this license allows unrestricted use and distribution, even commercially, by third parties. Please reply and provide explicit written permission to publish XXX under a CC BY license and complete the attached form.” Please upload the completed Content Permission Form or other proof of granted permissions as an "Other" file with your submission.  In the figure caption of the copyrighted figure, please include the following text: “Reprinted from [ref] under a CC BY license, with permission from [name of publisher], original copyright [original copyright year].” b. If you are unable to obtain permission from the original copyright holder to publish these figures under the CC BY 4.0 license or if the copyright holder’s requirements are incompatible with the CC BY 4.0 license, please either i) remove the figure or ii) supply a replacement figure that complies with the CC BY 4.0 license. Please check copyright information on all replacement figures and update the figure caption with source information. If applicable, please specify in the figure caption text when a figure is similar but not identical to the original image and is therefore for illustrative purposes only.

Reviewers' comments:

Reviewer's Responses to Questions

**Comments to the Author**

1. Is the manuscript technically sound, and do the data support the conclusions?

Reviewer #1: No

Reviewer #2: Yes

Reviewer #3: Yes

2. Has the statistical analysis been performed appropriately and rigorously? 

Reviewer #1: N/A

Reviewer #2: Yes

Reviewer #3: Yes

3. Have the authors made all data underlying the findings in their manuscript fully available?

Reviewer #1: Yes

Reviewer #2: Yes

Reviewer #3: Yes

4. Is the manuscript presented in an intelligible fashion and written in standard English?

Reviewer #1: Yes

Reviewer #2: Yes

Reviewer #3: Yes

5. Review Comments to the Author

**Reviewer #1: ** I appreciate the opportunity to review this manuscript and would like to extend my gratitude to both the authors and the editorial team for their efforts. The current research presents a taxonomic reassessment of the mnemosyne species group of the myrmicine genus Strumigenys, providing descriptions of three new species, commentary on previously described species, an identification key for all species within the group, and a workflow for the “virtual reconstruction” of broken specimens using micro-CT models.

This study represents a well-executed taxonomic analysis and will be a valuable addition to the Strumigenys literature. However, I believe it requires significant revisions to be suitable for publication in a taxonomy-oriented journal. More critically, I do not see how this study advances the broader fields of Biology or Taxonomy, as it presents two major issues that hinder its suitability for publication in this journal.

First, the authors argue that micro-CT models can help mitigate the risk of specimen breakage during handling and provide a method for reconstructing damaged specimens. However, this justification is problematic, as the specimen needing reconstruction was damaged during the micro-CT process itself. In this context, the “reconstruction workflow” appears to be a solution to a problem introduced by the study’s own methodology. Additionally, contrary to the authors’ claims, the micro-CT models did not contribute to the observation of difficult-to-see structures in the head, such as the dentition formula. These structures are more clearly visible in the high-resolution images, as evidenced by the difficulty in distinguishing smaller preapical teeth in the micro-CT model of S. marmorata compared to the corresponding high-resolution image.

Second, while I partly agree with the authors that digital surrogates could help alleviate problems associated with operational procedures (e.g. breaking specimens), it is important to highlight that it does not alleviate the issue of accessibility to physical type specimens housed in collections when intended to be used for revisionary studies, nor solves the issues of reproducibility or replicability in taxonomy. Since "digital objects do not represent anything, in either sense, but gain roles and capacities in their use in different social settings” (Boast & Enote, 2013), cybertypes would serve only as iconic representational artifacts (i.e. human-made concretizations of cognitive representations that carry aesthetic nonconceptual content by resemblance to the physical item), and, thus, would play the same role as 2D images. This means that they could not be used by other researchers interested in reevaluating species boundaries, because data (in this case, formalized statements of morphological features made by a researcher based on the physical specimen) would have to be created by documenting morphological features observed directly from the physical type specimen; otherwise, researchers would be only describing human-made concretizations, instead of material entities (like describing Dorian Grey based on Henrique Medina’s painting).

Given these concerns, I recommend rejecting this manuscript for publication in this journal and suggest that the authors consider submitting it to a taxonomy-focused journal instead.

Thank you once again for the opportunity to review this manuscript.

Boast, R., Enote, J. (2013). Virtual Repatriation: It Is Neither Virtual nor Repatriation. In: Heritage in the Context of Globalization. SpringerBriefs in Archaeology, vol 8. Springer, New York, NY. https://doi.org/10.1007/978-1-4614-6077-0_13

**Reviewer #2:**  Well drafted and presented

**Reviewer #3: ** After review of the manuscript PONE-D-25-13901, I believe the three species described, S. liui sp. nov., S. marmorata sp. nov., and S. rimdahli sp. nov., are new to science. The discovery of new members obviously increased our knowledge of the rare Strumigenys mnemosyne group. The results are worth to publish on the journal. However, there are numerous errors or inappropriate expressions in the manuscript. It needs to be published after major revisions.

1. Line 6: Change “Kadoorie Biological Sciences, Building” to “Kadoorie Biological Sciences Building”.

2. Lines 60-61: Change “singleton holotype specimens” to “singleton holotype specimen”.

3. Lines 89-91: This sentence “We provide an updated dichotomous key and distributional checklist and map of the whole S. mnenosyne group, and high-resolution images of all species, including the first for S. runa (Bolton, 2000)” used too many “and”, and the meaning of last part is not clear. I suggest the authors to use simple sentences to express clearly.

4. Lines 110-135: The authors used some complex abbreviation for the measurements “PrW, PetL, PetH, DPetW, DPetL, DPPetW, DDPetL”, why don’t you use the simple abbreviation “PW, PL, PH, DPW, DPL, PPW, PPL” as in Bolton (2000)? Because scientific report is trying to make thing simple, not the contrary.

5. Lines 157-179: The authors habitually used too many “we” in the whole paragraph and seems too colloquial. I suggest the authors change the expression of sentences and reduced the use of “we” frequently.

6. Lines 184-185: “Hita Garcia and colleagues (2025)” is not a standard literature citation, please change it to “Hita Garcia et al. (2025)”.

7. Line 240: This sentence “Dorsum and sides of the mesosoma declivity of propodeum” seems incorrect, please rewrite.

8. Line 262: Change “1 Holotype worker” to “Holotype worker”, because holotype is a single individual.

9. Lines 269-270: In “Nothing is known of the biology of S. daspleta but it is presumed to have been collected within leaf litter”, the subjective guess “but it is presumed to have been collected within leaf litter” is perilous and maybe cause a mistake. If the biological information of the species was not recorded in the original description, it is better keeping unknown until new information obtained in the future investigation.

10. Line 275: Table 1 short of an end line.

11. Line 289: Change “Physical specimens examined (n=1)” to “Physical specimen examined (n=1)”.

12. Line 290: Change “1 Holotype worker” to “Holotype worker”; change “Baihua Lin” to “Baihua Ling”.

13. Line 290: Change “N25.302, E.098.789” to “N25.302, E98.789”.

14. Line 302: Change “Holotype worker (n = 1)” to “Holotype worker”, “(n = 1)” is redundant.

15. Line 306: Change “Occiput” to “Posterior margin”, the correct name of this part is posterior margin in morphology.

16. Line 310: Change “in full face view” to “in full-face view”.

17. Line 314: Change “upper masticatory margin” to “Basal masticatory margin”.

18. Lines 314-315: Change “lower half” to “apical half”.

19. Line 319: Change “convex below eye” to “convex before eye”.

20. Lines 323-324: The meaning of “Mesosoma undifferentiated” is not clear. It is better to change the whole sentence “Mesosoma undifferentiated, lacking pleural sutures other than weakly impressed promesonotal suture not reaching dorsolateral margin in lateral view” to “Mesosoma lacking promesonotal suture on the dorsum, metanotal groove absent”.

21. Line 326: What is “Hair wheel”?

22. Line 334: This description “Petiolar spiracle located just below node on anterolateral region of peduncle” is wrong, please check and rewrite.

23. Line 341: About pilosity, the setae in Fig.3D-F are obviously shorter and stouter than in Fig.3A-C, why? It seems the shape of setae in Fig.3D-F distortion. Similar in Fig.4-8.

24. Line 346: Change “apically pointed setae” to “apically acute setae”.

25. Line 354: It is not clear whose “Tergite one”? please rewrite.

26. Line 359: Change “Mandibles and antennal segments dulled by dense microsculpture” to “Mandibles and antennal segments with dense micropuncture”. Sculpture is a general term for all kind of sculptures.

27. Line 362: Change “Overall body reddish-orange, appendages yellow. Spongiform tissue yellow-white” to “Body reddish orange, appendages yellow. Spongiform tissue light yellow”.

28. Line 385: Change “tolerance for higher elevations conditions” to “tolerance for higher elevation condition”.

29. Line 395: Change “In full face view” to “In full-face view”.

30. Lines 395-396: “in lateral view” is repeated.

31. Line 397: Change “Upper masticatory margin” to “Basal masticatory margin”.

32. Line 398: Change “lower half” to “apical half”.

33. Line 406: Change “1 Holotype worker” to “Holotype worker”.

34. Line 406: Change “22.26906 113.93088” to “N22.26906, E113.93088”.

35. Line 409: Change “22.51211 114.0928” to “N 22.51211, E114.0928”.

36. Line 428: Change “Holotype measurements (n = 1)” to “Holotype measurements”.

37. Line 435” Change “Occiput” to “Posterior margin”.

38. Line 438: Change “in full face view” to “in full-face view”.

39. Line 439: Change “occiput” to “posterior margin”.

40. Lines 441-442: Change “upper masticatory margin” to “basal masticatory margin”, change “lower half” to “apical half”.

41. Line 445: Change “single ommatidia” to “single ommatidium”.

42. Line 446: Change “convex below eye” to “convex before eye”.

43. Line 450: The meaning of “Mesosoma undifferentiated” is not clear. It is better to change the whole sentence “Mesosoma undifferentiated, lacking pleural sutures other than weakly impressed promesonotal suture not reaching dorsolateral margin in lateral view” to “Mesosoma lacking promesonotal suture on the dorsum, metanotal groove absent”.

44. Line 459-460: Anterior peduncle of petiole is much longer in Fig. 4A than in Fig. 4E, they are the same species, why? Meanwhile, anterior margin of postpetiole weakly concave in Fig.4C, in contrast, anterior margin of postpetiole straight in Fig.4F, why? Are they really belonging to the same species?

45. Line 462: “kidney bean shaped” is not a good description, please describe using geometry terms.

46. Line 462: Change “postpetiolar disc conspicuously concave” to “postpetiole conspicuously concave”.

47. Line 463: “parabolic arc” is not accurate, please rewrite.

48. Lines 481-482: Change “post-petiole” to “postpetiole”, change “onepair” to “one pair”.

49. Line 486: Change “meso- and metabasitarsi and tibiae” to “meso- and meta- basitarsi and tibiae”.

50. Lines 490-492: Change “Core body segments reddish-brown; appendages of similar but lighter colour; antennal segments brighter compared to body, considerably more orange; spongiform tissue of typical whitish yellow colouration” to “Body color reddish brown, appendages light reddish brown, spongiform tissue light yellow”.

51. Line 495: Change “micro-sculpture” to “micro-punctures”.

52. Lines 508-522: In this paragraph, the authors say “S. marmorata most closely resembles S. mnemosyne and S. rimdahli”. They have compared S. marmorata and S. mnemosyne carefully, but no difference mentioned between S. marmorata and S. rimdahli. Please supplement and improve.

53. Lines 528-529: Change “Strumigenys emmae, (Emery, 1890)” to “Strumigenys emmae (Emery, 1890)”.

54. Line 537: Change “Digital images of holotype (A) lateral, (B) dorsal, (C) full face view” to “Digital images of holotype (A) head, (B) lateral, (C) dorsal”.

55. Line 556: Change “1 Holotype worker” to “Holotype worker”.

56. Line 560: Change “22.42007 114.1829” to “N22.42007, E114.1829”, change the similar latitude and longitude in the whole paragraph from Line 558 to Line 580, and on Lines 603-610.

57. Line 622: Indices of gyne: CI 75.91–131.18, which showing a huge span and seems unusual, please check and make sure.

58. Line 625: Change “occipital border” to “posterior border”.

59. Lines 626-627: Change “upper masticatory margin” to “basal masticatory margin”, change “lower half” to “apical half.

60. Line 636: Change “Occipital border” to “Posterior margin”.

61. Lines 641: Change “mesoscutum anterior margin” to “anterior margin of mesoscutum”.

62. Line 655: The word “subclubate” seems wrong, please check and correct.

63. Line 659: Change “ventral petiole” to “ventral face of petiole”.

64. Lines 659-660: Change “First gastralastral tergite long, as long as petiole and postpetiole” to “First gastral tergite long, as long as petiole and postpetiole combined”.

65. Lines 680-688: Please make sure what is “microsculpture”, micro-puncture or micro-reticulation? Because “sculpture” is a general name for every kind of sculpture including puncture, reticulation, etc.

66. Line 690: Change “Overall body and appendages red-orange, spongiform tissue yellow-white” to “Body and appendages reddish orange, spongiform tissue light yellow”.

67. Line 726: Change “1 Holotype worker” to “Holotype worker”.

68. Line 753: Change “1 Holotype worker” to “Holotype worker”, change “19.236638 98.650739” to “N19.236638, E98.650739”.

69. Line 769: Change “Occiput” to “Posterior margin”.

70. Line 773: Change “occiput” to “posterior margin”.

71. Lines 775-776: Change “upper masticatory margin” to “basal masticatory margin”, change “lower half” to “apical half”.

72. Line 780: Change “single ommatidia” to “single ommatidium”.

73. Line 781: Change “below eye” to “before eye”.

74. Lines 785-786: The meaning of “Mesosoma undifferentiated” is not clear. It is better to change the whole sentence “Mesosoma undifferentiated, lacking pleural sutures other than weakly impressed promesonotal suture not reaching dorsolateral margin in lateral view” to “Mesosoma lacking promesonotal suture on the dorsum, metanotal groove absent”.

75. Lines 880 and 802: Change “post-petiole” to “postpetiole”.

76. Line 815: Change “fourc to five” to “four to five”.

77. Lines 824-826: Change “Core body segments reddish-brown; appendages of similar but lighter colour; antennal segments brighter compared to body, considerably more orange; spongiform tissue of typical whitish yellow colouration” to “Body color reddish brown, appendages light reddish brown, spongiform tissue whitish yellow”.

78. Line 829: Please make sure what kind of sculpture on antennae, coxae and basitarsi.

79. Line 838: Change “rapidly” to “easily”.

80. Line 843: Chang “in mnemosyne” to “in S. mnemosyne”.

81. Line 852: Change “of the Strumigenys rimdahli sp. nov.” to “of Strumigenys rimdahli sp. nov.”.

82. Line 853: Change “Digital images (A) lateral view, (B) dorsal view, (C) full face view” to “Digital images (A) head, (B) lateral, (C) dorsal”.

83. Line 865: Change “4.73356 116.87757 ± 50m” to “N4.73356, E116.87757”.

84. Line 875: Change “Digital images (A) lateral view, (B) dorsal view, (C) full face view” to “Digital images (A) head, (B) lateral, (C) dorsal”, meanwhile, rearrange the images of Fig.9 in the order: (A) head, (B) lateral, (C) dorsal.

85. Line 878” Change “Fig 2G–I” to “Fig. 2G–I”.

86. Line 884: Change “highly concave” to “deeply concave”.

87. Line 889: Change “1 Holotype worker” to “Holotype worker".

88. Lines 907-908: Change “Key to the Strumigenys mnemosyne group (adapted from Bolton 2000; Southeast Asia Strumigenys key)” to “Key to species of the Strumigenys mnemosyne group based on worker caste”.

89. Line 909: Change “This key branches from couplet 84 of Bolton (2000), ending at couplet 87” to “(Adapted from the Southeast Asia Strumigenys key of Bolton, 2000, branching from couplet 84 and ending at couplet 87)”.

90. Line 911: Because the opposite sentence of antithetical couplet 1 short of “Basigastral costulae” character, please delete “Basigastral costulae absent” here.

91. Line 917: Change “anterodorsally . . . . . 4” to “anterodorsally . . . . . 5”.

92. Lines 935-939: It seems the characters the authors provided to separate the two species are hard to use. Please find more other clear characters to separate the two species.

93. Line 950: Change “The mnemosyne group” to the mnemosyne group

94. Line 952: Change “absence of spines” to “absence of propodeal spines”.

95. Line 954: Change “the mnemosyne group” to “the S. mnemosyne group”.

6. PLOS authors have the option to publish the peer review history of their article (what does this mean? ). If published, this will include your full peer review and any attached files.

**Do you want your identity to be public for this peer review?** For information about this choice, including consent withdrawal, please see our Privacy Policy .

Reviewer #1: **Yes: ** Thiago Sanches Ranzani da Silva

Reviewer #2: **Yes: ** Himender Bharti

Reviewer #3: No

---

## [Author Response · Author response to Decision Letter 1]

31 Jul 2025

Response to the editor

We would like to thank the editor, and the three reviewers for their thoughts and comments on our manuscript. Nearly all comments from reviewer 2 and 3 have been included into the manuscript. Where their comments have not been included in the manuscript, a rebuttal has been provided. Concerning concerns raised by reviewer 1, their comments are based more in philosophy, are somewhat irrelevant to our paper, and based on unscientific presumptions which are substantiated by the current literature. Further, they seem to be confused by the aims and scope of PLOS One, suggesting instead that we should submit to a taxonomy specific journal, but provide no justification for this. Therefore, we believe that their comments are mostly irrelevant to our paper.

Our responses to the reviewers can be found in red.

Reviewer 1 response

Reviewer #1: I appreciate the opportunity to review this manuscript and would like to extend my gratitude to both the authors and the editorial team for their efforts. The current research presents a taxonomic reassessment of the mnemosyne species group of the myrmicine genus Strumigenys, providing descriptions of three new species, commentary on previously described species, an identification key for all species within the group, and a workflow for the “virtual reconstruction” of broken specimens using micro-CT models.

This study represents a well-executed taxonomic analysis and will be a valuable addition to the Strumigenys literature. However, I believe it requires significant revisions to be suitable for publication in a taxonomy-oriented journal. More critically, I do not see how this study advances the broader fields of Biology or Taxonomy, as it presents two major issues that hinder its suitability for publication in this journal.

First, the authors argue that micro-CT models can help mitigate the risk of specimen breakage during handling and provide a method for reconstructing damaged specimens. However, this justification is problematic, as the specimen needing reconstruction was damaged during the micro-CT process itself. In this context, the “reconstruction workflow” appears to be a solution to a problem introduced by the study’s own methodology. Additionally, contrary to the authors’ claims, the micro-CT models did not contribute to the observation of difficult-to-see structures in the head, such as the dentition formula. These structures are more clearly visible in the high-resolution images, as evidenced by the difficulty in distinguishing smaller preapical teeth in the micro-CT model of S. marmorata compared to the corresponding high-resolution image.

Second, while I partly agree with the authors that digital surrogates could help alleviate problems associated with operational procedures (e.g. breaking specimens), it is important to highlight that it does not alleviate the issue of accessibility to physical type specimens housed in collections when intended to be used for revisionary studies, nor solves the issues of reproducibility or replicability in taxonomy. Since "digital objects do not represent anything, in either sense, but gain roles and capacities in their use in different social settings” (Boast & Enote, 2013), cybertypes would serve only as iconic representational artifacts (i.e. human-made concretizations of cognitive representations that carry aesthetic nonconceptual content by resemblance to the physical item), and, thus, would play the same role as 2D images. This means that they could not be used by other researchers interested in reevaluating species boundaries, because data (in this case, formalized statements of morphological features made by a researcher based on the physical specimen) would have to be created by documenting morphological features observed directly from the physical type specimen; otherwise, researchers would be only describing human-made concretizations, instead of material entities (like describing Dorian Grey based on Henrique Medina’s painting).

Given these concerns, I recommend rejecting this manuscript for publication in this journal and suggest that the authors consider submitting it to a taxonomy-focused journal instead.

Thank you once again for the opportunity to review this manuscript.

Boast, R., Enote, J. (2013). Virtual Repatriation: It Is Neither Virtual nor Repatriation. In: Heritage in the Context of Globalization. SpringerBriefs in Archaeology, vol 8. Springer, New York, NY. https://doi.org/10.1007/978-1-4614-6077-0_13

Response: We would like to thank reviewer 1 for their thoughts on our manuscript, however we think the arguments presented are somewhat irrelevant here, overly philosophical, and do not reflect the scientific content presented in our study. Below we provide a rebuttal to the ideas and opinions they have presented.

Firstly, from our understanding, it seems that the reviewer is confused by the aims and scope of PLOS One. Reviewer 1 has stated ‘I do not see how this study advances the broader fields of Biology or Taxonomy’. On the PLOS One webpage Journal Information (https://tinyurl.com/yc42c2p3) it states that ‘… we evaluate research on the basis of scientific validity, strong methodology, and high ethical standards—not perceived significance’. As also put by reviewer 1‘This study represents a well-executed taxonomic analysis and will be a valuable addition to the Strumigenys literature’, this rather contradictory statement implies our paper has met PLOS One standards by being both scientifically valid and methodology in terms of methodology. Despite those statements, the reviewer suggests our paper should be revised and submitted to a taxonomic journal (without reasoning). However, there is no greater generality or novelty required for PLOS ONE, and as inadvertently stated by the reviewer themselves, our paper has met the journals requirements. Nevertheless, we argue below that the points they raise in their review can largely be discarded as those are either the results of a misunderstanding (to which we bring precisions in this new version) or irrelevant to our study.

During further dialogue with coauthors, it was realised that the specimen broke not during the CT-scanning process itself, but upon opening the specimen transportation box before any other manipulation of the specimen. This has been rectified within the manuscript as “The paratype specimen broke into three pieces whilst opening the specimen transportation box (Fig 5).”. Despite this, even if the specimen was broken because of our need to conduct the CT-scanning, it does not necessarily mean a problem arises because of the methodology used. Accidents may happen during any protocol, occurring for any reason, and not be fundamentally derived from the methodology itself. In our case, just because our broken specimens were destined for CT scanning (with the potential for virtual repair post scanning), does not equate to CT scanning introducing a problem. It just so happens an accident occurred prior to the application of a method that provided a solution to the problem. A similar accident could have happened for a traditional taxonomic revision for gaining new observations, or placing the specimen in a new collection.

The suggestion that the CT scanning images did not contribute to the observation of difficult to see structures is incorrect. The lead author was not able to discern the full dentition formula in any of the digital images, nor with a microscope at 150x magnification. Access to a 3D model was therefore invaluable in distinguishing the preapical teeth. It is not clear how the reviewer came to their conclusion as they do not have access to digital images of the apical and pre-apical dentition for comparison. Digital images of the pre-apical and apical dentition were not included because of the poor image resolution.

The use of CT scanning for taxonomic purposes has been in use now for over a decade. Virtual models of zoological specimens have enabled the observation of challenging to see morphological structures (Hita Garcia et al., 2017; Hita Garcia et al., 2019; Hita Garcia et al., 2025). Unless a specimen is physically dissected, which is not always possible, even previously hidden internal structures can be observed through virtual dissection (Richter et al., 2021; Aibekova et al., 2025). From our understanding, the reviewer has argued that morphological characters, observed on 3D models do not represent anything, and are merely human concretions. We believe that this is wrong, with this argument based more in philosophy rather than hard evidence. Numerous previously studies have utilised and incorporated CT scanning models into their character examinations. The reviewer also suggests cybertypes are meaningless given that they represent ‘nonconceptual content by resemblance to the physical item … and, thus, would play the same role as 2D images’. The notion that a character does not exist because it is only represented in virtual form is overly philosophical and unsubstantiated. 3D models have all morphological characters as the physical specimen, corresponding directly to the physical specimen rather than simply resembling it (Hita Garcia et al., 2017). Equating a 3D model to a 2D digital image is also highly dubious. With 3D models, one can dissect, rotate and zoom into a model to examine a specimen’s morphology, often allowing to perform measurements which are all impossible for digital images.

Reviewer 1 also states ‘it is important to highlight that it does not alleviate the issue of accessibility to physical type specimens housed in collections when intended to be used for revisionary studies, nor solves the issues of reproducibility or replicability in taxonomy’. We believe that CT scanned type species do indeed help to alleviate the issue regarding accessibility to type specimens as 3D models provide a powerful and flexible way to visualise specimens, with much greater accessibility than physical specimens as researchers only require a computer connected to the internet. We do not suggest 3D models should replace physical type specimens, nor do we argue this in the paper. But 3D models, surely offer researchers new alternatives and powerful way to access, view, and visualise type material. Further, the reviewer has referenced a single study referring to the repatriation of archaeological and historical artifacts of societal and cultural significance via 3D scans (Boast & Enote 2013), arguing that the artefacts religious and cultural value are not translated via 3D modelling. We believe that this opinion is not applicable to 3D scans of insect specimens, which have low cultural or to any religious value. Further, the issue of repatriation here is irrelevant, considering all physical type specimens and non-type specimens will be deposited within publicly accessible museums in the region where those specimens have been collected.

References

Aibekova L, Richter A, Beutel RG, van de Kamp T, Economo EP, Griebenow Z, Boudinot B. (2025) The Mesosoma of Protanilla (Leptanillinae) and the Groundplan of the Formicidae (Hymenoptera), Journal of Morphology, 286(7). https://doi.org/10.1002/jmor.70064.

Boast R, Enote J (2013). Virtual Repatriation: It Is Neither Virtual nor Repatriation. In: Heritage in the Context of Globalization. SpringerBriefs in Archaeology, vol 8. Springer, New York, NY. https://doi.org/10.1007/978-1-4614-6077-0_13

Hita Garcia F, Fischer G, Liu C, Audisio TL, Economo EP (2017) Next-generation morphological character discovery and evaluation: an X-ray micro-CT enhanced revision of the ant genus Zasphinctus Wheeler (Hymenoptera, Formicidae, Dorylinae) in the Afrotropics. ZooKeys 693: 33-93. https://doi.org/10.3897/zookeys.693.13012

Hita Garcia F, Gómez K, Keller RA, Schurian B, Economo EP (2025) A never-ending story: updated 3D cyber-taxonomic revision of the ant genus Zasphinctus Wheeler (Hymenoptera, Formicidae, Dorylinae) for the Afrotropical region. ZooKeys 1223: 1-55. https://doi.org/10.3897/zookeys.1223.131238

Hita-Garcia F, Lieberman Z, Audisio TL, Liu C, Economo EP, Revision of the Highly Specialized Ant Genus Discothyrea (Hymenoptera: Formicidae) in the Afrotropics with X-Ray Microtomography and 3D Cybertaxonomy, Insect Systematics and Diversity, Volume 3, Issue 6, November 2019, 5, https://doi.org/10.1093/isd/ixz015

Richter A, Hita Garcia F, Keller RA, Billen J, Katzke J, Boudinot BE, Economo EP, Beutel RG (2021) The head anatomy of Protanilla lini (Hymenoptera: Formicidae: Leptanillinae), with a hypothesis of their mandibular movement, Myrmecological News, 31, pp. 85–114.

Reviewer 2 response

Reviewer #2: Well drafted and presented

A revision of the Strumigenys mnemosyne group has been nicely attempted by the authors. It would be better if images of the species are referred in key.

Many thanks for reviewing our manuscript. As suggested by the reviewer, we have now incorporated figure citations into our dichotomous key.

Reviewer 3

PONE-D-25-13901 Reviewing conclusion 20250607 by Zhenghui Xu

After review of the manuscript PONE-D-25-13901, I believe the three species described, S. liui sp. nov., S. marmorata sp. nov., and S. rimdahli sp. nov., are new to science. The discovery of new members obviously increased our knowledge of the rare Strumigenys mnemosyne group. The results are worth to publish on the journal. However, there are numerous errors or inappropriate expressions in the manuscript. It needs to be published after major revisions.

1. Line 6: Change “Kadoorie Biological Sciences, Building” to “Kadoorie Biological Sciences Building”.

Thank you for highlighting this. Your recommendation has been added to the manuscript at the respective location.

2. Lines 60-61: Change “singleton holotype specimens” to “singleton holotype specimen”.

Thank you for highlighting this, however, your alteration is incorrect, as we are referring to multiple holotype specimens for multiple distinct species. We have therefore retained this in the manuscript at this respective location.

3. Lines 89-91: This sentence “We provide an updated dichotomous key and distributional checklist and map of the whole S. mnenosyne group, and high-resolution images of all species, including the first for S. runa (Bolton, 2000)” used too many “and”, and the meaning of last part is not clear. I suggest the authors to use simple sentences to express clearly.

Thank you for highlighting this, we have changed this to ‘We provide an updated dichotomous key, distributional checklist and map of the whole S. mnenosyne group, as well as high-resolution images of all species, including the first for S. runa (Bolton, 2000).’

4. Lines 110-135: The authors used some complex abbreviation for the measurements “PrW, PetL, PetH, DPetW, DPetL, DPPetW, DDPetL”, why don’t you use the simple abbreviation “PW, PL, PH, DPW, DPL, PPW, PPL” as in Bolton (2000)? Because scientific report is trying to make thing simple, not the contrary.

Thank you for this comment. Other than PW, Bolton 2000 did not diagnose these measurement types. Nevertheless, I have simplified these acronyms by changing PrW to PW, and removed all mention of ‘Pet’ from the acronym. I have kept the D (=dorsal) to ensure that the readers know these measurements refer to dorsal measurements. The new acroynms are as follows: PrW = PW; PetL = PL; PetH = PH; DPetW = DPW; DPetL = DPL; DPPetW = DPPW; DDPetL = DPPL.

5. Lines 157-179: The authors habitually used too many “we” in the whole paragraph and seems too colloquial. I suggest the authors change the expression of sentences and reduced the use of “we” frequently.

Thank you for this comment. The word ‘we’ has been removed from the entire manuscript other than within the acknowledgement sections. The structure of various sentences has been altered to reflect the removal of this word.

6. Lines 184-185: “Hita Garcia and colleagues (2025)” is not a standard literature citation, please change it to “Hita Garcia et al. (2025)”.

Thank you for highlighting this. Your recommendation has been added to the manuscript at the respective location.

7. Line 240: This sentence “Dorsum and sides of the mesosoma declivity of propodeum” seems incorrect, please rewrite.

Thank you for highlighting this; this has been changed to ‘Dorsum and sides of the mesosoma,

---

## [Decision Letter · Decision Letter 1]

21 Aug 2025

A revision of the rare Strumigenys mnemosyne (Formicidae; Myrmicinae) group using micro-CT scanning, with the description of three new species, and the virtual repair of a broken paratype.

PONE-D-25-13901R1

Dear Dr. Hamer,

Thank you for revising the manuscript. I have reviewed it and confirmed that all the queries and suggestions from the first round of review have been adequately addressed, and the overall quality of the manuscript has improved. I appreciate the authors’ efforts in revising the manuscript with great precision. We’re pleased to inform you that your manuscript has been judged scientifically suitable for publication and will be formally accepted for publication once it meets all outstanding technical requirements. 

Kind regards,

A.P. Ranjith

Academic Editor

PLOS ONE

Additional Editor Comments (optional):

Reviewers' comments:

Reviewer's Responses to Questions

**Comments to the Author**

1. If the authors have adequately addressed your comments raised in a previous round of review and you feel that this manuscript is now acceptable for publication, you may indicate that here to bypass the “Comments to the Author” section, enter your conflict of interest statement in the “Confidential to Editor” section, and submit your "Accept" recommendation.

Reviewer #3: All comments have been addressed

2. Is the manuscript technically sound, and do the data support the conclusions?

Reviewer #3: Yes

3. Has the statistical analysis been performed appropriately and rigorously? 

Reviewer #3: Yes

4. Have the authors made all data underlying the findings in their manuscript fully available?

Reviewer #3: Yes

5. Is the manuscript presented in an intelligible fashion and written in standard English?

Reviewer #3: Yes

6. Review Comments to the Author

Reviewer #3: After a careful read of the Response to Reviewers and the revised manuscript, I noticed that the authors have fully revised and improved the manuscript. I agree to publish the paper on the journal.

7. PLOS authors have the option to publish the peer review history of their article (what does this mean? ). If published, this will include your full peer review and any attached files.

**Do you want your identity to be public for this peer review?** For information about this choice, including consent withdrawal, please see our Privacy Policy .

Reviewer #3: No

---

## [Editor Report · Acceptance letter]

PONE-D-25-13901R1

PLOS ONE

Dear Dr. Hamer,

I'm pleased to inform you that your manuscript has been deemed suitable for publication in PLOS ONE. Congratulations! Your manuscript is now being handed over to our production team.

Kind regards,

on behalf of

Dr. A.P. Ranjith

Academic Editor

PLOS ONE